# A Provably Efficient Sample Collection Strategy for Reinforcement Learning

**Jean Tarbouriech**
Facebook AI Research Paris & Inria Lille
jean.tarbouriech@gmail.com

**Matteo Pirotta**
Facebook AI Research Paris
pirotta@fb.com

**Michal Valko**
DeepMind Paris
valkom@deepmind.com

**Alessandro Lazaric**
Facebook AI Research Paris
lazaric@fb.com

## Abstract

One of the challenges in *online* reinforcement learning (RL) is that the agent needs to trade off the exploration of the environment and the exploitation of the samples to optimize its behavior. Whether we optimize for regret, sample complexity, state-space coverage or model estimation, we need to strike a different exploration-exploitation trade-off. In this paper, we propose to tackle the exploration-exploitation problem following a decoupled approach composed of: **1)** An "objective-specific" algorithm that (adaptively) prescribes *how many* samples to collect *at which* states, as if it has access to a generative model (i.e., a simulator of the environment); **2)** An "objective-agnostic" sample collection exploration strategy responsible for generating the prescribed samples as fast as possible. Building on recent methods for exploration in the stochastic shortest path problem, we first provide an algorithm that, given as input the number of samples $b(s, a)$ needed in each state-action pair, requires $\widetilde{O}\big(BD + D^{3/2}S^2A\big)$ time steps to collect the $B = \sum_{s,a} b(s, a)$ desired samples, in any unknown communicating MDP with $S$ states, $A$ actions and diameter $D$. Then we show how this general-purpose exploration algorithm can be paired with "objective-specific" strategies that prescribe the sample requirements to tackle a variety of settings — e.g., model estimation, sparse reward discovery, goal-free cost-free exploration in communicating MDPs — for which we obtain improved or novel sample complexity guarantees.

## 1 Introduction

One of the challenges in *online* reinforcement learning (RL) is that the agent needs to trade off the exploration of the environment and the exploitation of the samples to optimize its behavior. Whenever the agent needs to gather information about a specific region of the Markov decision process (MDP), it must plan for a policy to reach the desired states, despite not having exact knowledge of the environment dynamics. This makes solving the exploration-exploitation problem in RL highly non-trivial and it requires designing a specific strategy depending on the learning objective, such as PAC-MDP learning [e.g., 13, 47, 59], regret minimization [e.g., 28, 6, 29, 66] or pure exploration [e.g., 30, 31, 39, 63, 64].

A simpler scenario considered in the literature is to assume access to a *generative model* or *sampling oracle* ($\mathcal{SO}$) [33]. Given any state-action pair $(s, a)$, the $\mathcal{SO}$ returns a next state $s'$ drawn from the transition probability $p(\cdot|s, a)$ and a reward $r(s, a)$. In this case, it is possible to focus exclusively on where and how many samples to collect, while disregarding the problem of finding a suitable policy

35th Conference on Neural Information Processing Systems (NeurIPS 2021).

to obtain them. For instance, an $\mathcal{SO}$ can be used to obtain samples from the environment, which are combined with dynamic programming techniques to compute an $\varepsilon$-optimal policy. $\mathcal{SO}$-based algorithms can be as simple as prescribing the same amount of samples from each state-action pair [e.g. 35, 33, 5, 16, 46, 1, 38] or they may adaptively change the sample requirements on different state-action pairs [e.g. 15, 58, 62]. An $\mathcal{SO}$ is also used in Monte-Carlo planning [49, 25, 7] which focuses on computing the optimal action at the current state by optimizing over rollout trajectories sampled from the $\mathcal{SO}$. Finally, in multi-armed bandit [37], there are cases where each arm corresponds to a state (or state-action), and "pulling" an arm translates into a call to an $\mathcal{SO}$ (see e.g., the pure exploration setting of [51]). Unfortunately, while an $\mathcal{SO}$ may be available in domains such as simulated robotics and computer games, this is not the case in the more general *online RL* setting.

In this paper we tackle the exploration-exploitation problem in online RL by drawing inspiration from the $\mathcal{SO}$ assumption. Specifically, we define an approach that is decoupled in two parts: **1)** an "objective-specific" algorithm that assumes access to an $\mathcal{SO}$ that (adaptively) prescribes the samples needed to achieve the learning objective of interest, and **2)** an "objective-agnostic" algorithm that takes on the exploration challenge of collecting the samples requested by the $\mathcal{SO}$-based algorithm as quickly as possible.[1] Our main contributions can be summarized as follows:

- We define the sample complexity of the objective-agnostic algorithm as the number of (online) steps needed to satisfy the prescribed sampling requirements. Leveraging recent techniques on exploration in the stochastic shortest path (SSP) problem [45, 50], we propose GOSPRL (Goal-based Optimistic Sampling Procedure for RL), a conceptually simple and flexible exploration algorithm that learns how to "generate" the samples requested by any $\mathcal{SO}$-based algorithm and we derive bounds on its sample complexity.

- Leveraging the generality of our approach, we combine GOSPRL with problem-specific $\mathcal{SO}$-based algorithms and readily obtain online RL algorithms in difficult exploration problems. While in general our decoupled approach may be suboptimal compared to exploration strategies designed to solve one specific problem, we obtain sample complexity guarantees that are on par or better than state-of-the-art algorithms in a range of problems. **1)** GOSPRL solves the problem of sparse reward discovery in $\widetilde{O}\big(D^{3/2}S^2A\big)$ time steps, which improves the dependency on the diameter $D$ w.r.t. a reward-free variant of UCRL2B [28, 22], as well as on $S$ and $A$ w.r.t. a MAXENT-type approach [26, 17]. **2)** GOSPRL improves over the method of [54] for model estimation, by removing their ergodicity assumption as well as achieving better sample complexity. **3)** GOSPRL provably tackles the problem of goal-free cost-free exploration, for which no specific strategy is available.

- We report numerical simulations supporting our theoretical findings and showing that pairing GOSPRL with $\mathcal{SO}$-based algorithms outperforms both heuristic and theoretically grounded baselines in various problems.

**Related work.** While to the best of our knowledge no other work directly addresses the problem of simulating an $\mathcal{SO}$, a number of approaches are related to it. The problem solved by GOSPRL can be seen as a *reward-free* exploration problem, since it is not driven by any external reward but by the objective of covering the state space to quickly meet the sampling requirements. Standard exploration-exploitation algorithms, such as UCRL2 [28] in the undiscounted setting or RMAX [13] in the discounted one, implicitly encourage exploration to specific areas of the state-action space that are not estimated accurately enough. The objective of covering the state space is also studied in [26, 17] with a Frank-Wolfe approach that optimizes a smooth aggregate function of the state visitations.

Recent works on reward-free exploration (RFE) in the finite-horizon setting [e.g., 30, 31, 39, 64] provide sufficient exploration so that an $\varepsilon$-optimal policy for *any* reward function can be computed. Our proposed solution shares high-level algorithmic principles with RFE approaches which incentivize the agent to visit insufficiently visited states via intrinsic reward. Nonetheless, our contribution significantly differs from existing RFE literature in two dimensions: **1)** While we study the performance of GOSPRL in one goal-conditioned RFE problem (Sect. 4.3), our framework is much broader and it allows us to tackle a wider and diverse set of problems (Sect. 4 and App. I); **2)** Our setting is *horizon-agnostic* and *reset-free*, which prevents from directly using any method or technical analysis in RFE designed for problems with an imposed planning horizon (e.g., finite-horizon or discounted).

Finally, GOSPRL draws inspiration from the SSP formalism and solutions of [50, 45], but our approach critically differs from these works in three main ways: **1)** we are interested in sample

---

[1]Alternatively, we can view it as a general approach to take any $\mathcal{SO}$-based algorithm and convert it into an online RL algorithm.

complexity guarantees rather than a regret analysis; **2)** we consider requirements (i.e., goals to sample) that vary throughout the learning process, instead of an SSP problem with fixed goal state and cost function; **3)** we show how GOSPRL can serve as a sample collection component to tackle various learning problems other than regret minimization.

## 2  Problem Definition

We consider a finite and *reset-free* MDP [43] $M := \langle \mathcal{S}, \mathcal{A}, p, r, s_0 \rangle$, with $S := |\mathcal{S}|$ states, $A := |\mathcal{A}|$ actions and an arbitrary starting state $s_0 \in \mathcal{S}$. Calling an $\mathcal{SO}$ in any state-action pair $(s, a)$ leads to two outcomes: a next state sampled from the transition probability distribution $p(\cdot|s, a) \in \Delta(\mathcal{S})$, and a reward $r(s, a) \in \mathbb{R}$. A stationary deterministic policy is a mapping $\pi : \mathcal{S} \to \mathcal{A}$ from states to actions and we denote by $\Pi^{\text{SD}}$ the set of all such policies. For any policy $\pi$ and pair of states $(s, s')$, let $\tau_\pi(s \to s')$ be the (possibly infinite) hitting time from $s$ to $s'$ when executing $\pi$, i.e., $\tau_\pi(s \to s') := \inf\{t \geq 0 : s_{t+1} = s' \,|\, s_1 = s, \pi\}$, where $s_t$ is the state visited at time step $t$. We introduce

$$D_{ss'} := \min_{\pi \in \Pi^{\text{SD}}} \mathbb{E}[\tau_\pi(s \to s')], \qquad D_{s'} := \max_{s \in \mathcal{S} \setminus \{s'\}} D_{ss'}, \qquad D := \max_{s' \in \mathcal{S}} D_{s'},$$

where $D_{ss'}$ is the shortest-path distance between $s$ and $s'$, $D_{s'}$ is the SSP-diameter of $s'$ [50] and $D$ is the MDP diameter [28].

We now formalize the problem of simulating an $\mathcal{SO}$ (i.e., to generate the samples prescribed by an $\mathcal{SO}$-based algorithm). At each time step $t \geq 1$ the agent receives a function $b_t : \mathcal{S} \times \mathcal{A} \to \mathbb{N}$, where $b_t(s, a)$ defines the total number of samples that need to be collected at $(s, a)$ by time step $t$. We consider that $(b_t)_{t \geq 1}$ is an arbitrary sequence with each $b_t$ measurable w.r.t. the filtration up to time $t$ (i.e., it may depend on the samples observed so far).[2] We focus on the objective of designing an *online algorithm* that minimizes the time required to collect the prescribed samples. Since the environment is initially unknown, we need to trade off between exploring states and actions to improve estimates of the dynamics and exploiting current estimates to collect the required samples as quickly as possible. We formally define the performance metric as follows.

**Definition 1.** *For any state-action pair, we denote by $N_t(s, a) := \sum_{i=1}^{t} \mathbb{1}_{\{(s_i, a_i) = (s, a)\}}$ the number of visits to state $s$ and action $a$ up to (and including) time step $t$. Given a sampling requirement sequence $b := (b_t)_{t \geq 1}$ with $b_t : \mathcal{S} \times \mathcal{A} \to \mathbb{N}$ and a confidence level $\delta \in (0, 1)$, we define the sample complexity of a learning algorithm $\mathfrak{A}$ as*

$$\mathcal{C}(\mathfrak{A}, b, \delta) := \min\{t > 0 : \mathbb{P}(\forall (s, a) \in \mathcal{S} \times \mathcal{A}, \ N_t(s, a) \geq b_t(s, a)) \geq 1 - \delta\}.$$

With no additional condition, it is trivial to define problems such that $\mathcal{C}(\mathfrak{A}, b, \delta) = +\infty$ for any algorithm. To avoid this case, we introduce the following assumptions.

**Assumption 1.** *The MDP $M$ is communicating with a finite and unknown diameter $D < +\infty$.*

**Assumption 2.** *There exist an unknown and bounded function $\bar{b} : \mathcal{S} \times \mathcal{A} \to \mathbb{N}$ such that the sequence $(b_t)_{t \geq 1}$ verifies: $\forall t \geq 1$, $\forall (s, a) \in \mathcal{S} \times \mathcal{A}$, $b_t(s, a) \leq \bar{b}(s, a)$.*

Asm. 1 guarantees that whatever state needs to be sampled, there exists at least one policy that can reach it in finite time almost-surely (notice that it is considerably weaker than the ergodicity assumption (App. J) often used in online RL, see e.g., [60, 40, 24]). Asm. 2 ensures that the sequence of sampling requirements does not diverge and can thus be fulfilled in finite time. These assumptions guarantee that the problem in Def. 1 is well-posed and the sample complexity is bounded.

A variety of problems can be cast under our decoupled approach, in the sense that they can be tackled by solving the problem of Def. 1 under a specific instantiation of the sampling requirement sequence $(b_t)_{t \geq 1}$. For instance, consider the problem of covering the state-action space (e.g., to discover a hidden sparse reward), then the requirement is immediately defined as $b_t(s, a) = 1$. In Sect. 4 and App. I, we review problems where defining $b_t$ can be as simple as computing the sufficient number of samples needed to reach a certain level of accuracy in estimating a quantity of interest (e.g., model estimation) or can be directly extracted from existing literature (e.g., $\varepsilon$-optimal policy learning).

We now provide a simple worst-case lower bound on the sample complexity (details in App. D).

---

[2] Allowing adaptive sampling requirements enables to pair GOSPRL with $\mathcal{SO}$-based algorithms that adjust their requirements *online* as samples are being generated (see e.g., Sect. 4.2).

**Algorithm 1** GOSPRL Algorithm
___

**Input:** sampling requirement sequence $(b_t)_{t\geq1}$ with $b_t : \mathcal{S} \times \mathcal{A} \to \mathbb{N}$ revealed at time $t$ (or anytime before).
**Initialize:** Set $\mathcal{G}_1 := \{s \in \mathcal{S} : \exists a \in \mathcal{A}, b_1(s,a) > 0\}$, time step $t := 1$, counters $N_1(s,a) := 0$, attempt index $k := 1$ and attempt counters $U_1(s,a) := 0, \nu_1(s,a) := 0$.
**while** $\mathcal{G}_k$ is not empty **do**
    Define the SSP problem $M_k$ with goal states $\mathcal{G}_k$, and compute its optimistic shortest-path policy $\widetilde{\pi}_k$.
    Set flag = True and counter $\nu_k(s,a) := 0$.
    **while** flag **do**
        Execute action $a_t := \widetilde{\pi}_k(s_t)$ and observe next state $s_{t+1} \sim p(\cdot|s_t, a_t)$.
        Increment counters $\nu_k(s_t, a_t)$ and $N_t(s_t, a_t)$.
        **if** $s_{t+1} \in \mathcal{G}_k$ or $\nu_k(s_t, a_t) > \{U_k(s_t, a_t) \vee 1\}$ **then**
            Set flag = False.
        **end if**
        Set $t += 1$.
    **end while**
    **if** $s_t \in \mathcal{G}_k$ **then**
        Execute an action $a$ s.t. $N_t(s_t, a) < b_t(s_t, a)$, observe next state $s_{t+1} \sim p(\cdot|s_t, a)$ and set $t += 1$.
    **end if**
    Set $U_{k+1}(s,a) := U_k(s,a) + \nu_k(s,a)$, $k += 1$.
    Update the set of goal states $\mathcal{G}_k := \{s \in \mathcal{S} : \exists a \in \mathcal{A}, N_{t-1}(s,a) < b_{t-1}(s,a)\}$.
**end while**
___

**Lemma 1.** *For any $S \geq 1$, there exists an MDP with $S$ states satisfying Asm. 1 such that for any sampling requirement $b : \mathcal{S} \to \mathbb{N}$ satisfying Asm. 2,*

$$\min_{\mathfrak{A}} \mathcal{C}\left(\mathfrak{A}, b, \tfrac{1}{2}\right) = \Omega\Big(\sum_{s \in \mathcal{S}} D_s b(s)\Big).$$

Lem. 1 shows that the (possibly non-stationary) policy minimizing the time to collect all samples requires $\Omega\big(\sum_s D_s b(s)\big)$ time steps in a worst-case MDP. We also notice that when the total sampling requirement $B$ is concentrated on the state $\overline{s}$ for which $D_{\overline{s}} = D$ (i.e., $b(s') = 0, \forall s' \neq \overline{s}$), the previous bound reduces to $\Omega(BD)$.

## 3 Online Learning for $\mathcal{SO}$ Simulation

We now introduce our algorithm for the problem in Def. 1, bound its sample complexity and discuss several extensions.

### 3.1 The GOSPRL Algorithm

In Alg. 1 we outline GOSPRL (*Goal-based Optimistic Sampling Procedure for Reinforcement Learning*). At each time step $t$, GOSPRL receives a sampling requirement $b_t : \mathcal{S} \times \mathcal{A} \to \mathbb{N}$. The algorithm relies on the principle of optimism in the face of uncertainty and proceeds through *attempts* to collect relevant samples. We index the attempts by $k = 1, 2, \dots$ and denote by $t_k$ the time step at the start of attempt $k$ and by $U_k := N_{t_k-1}$ the number of samples available at the start of attempt $k$. At each attempt, GOSPRL goes through the following steps: **1)** Cast the under-sampled states as goal states and define an associated unit-cost multi-goal SSP instance (with unknown transitions); **2)** Compute an optimistic shortest-path policy; **3)** Execute the policy until either a goal state is reached or a stopping condition is satisfied; **4)** If a sought-after goal state denoted by $g$ has been reached, execute an under-sampled action (i.e., an action $a$ such that $N_t(g,a) < b_t(g,a)$). The algorithm ends when the sampling requirements are met, i.e., at the first time $t \geq 1$ where $N_t(s,a) \geq b_t(s,a)$ for all $(s,a)$.

**Step 1.** At any attempt $k$ we begin by defining the set of all under-sampled states

$$\mathcal{G}_k := \{s \in \mathcal{S} : \exists a \in \mathcal{A}, N_{t_k-1}(s,a) < b_{t_k-1}(s,a)\}.$$

We then cast the sample collection problem as a goal-reaching objective, by constructing a multi-goal SSP problem [9] denoted by $M_k := \langle \mathcal{S}_k, \mathcal{A}, p_k, c_k, \mathcal{G}_k \rangle$, with:[3]

___

[3]If the current state $s_{t_k}$ is under-sampled (i.e., $s_{t_k} \in \mathcal{G}_k$), we duplicate the state and consider it to be both a goal state in $\mathcal{G}_k$ and a non-goal state from which the attempt $k$ starts (and whose outgoing dynamics are the same as those of $s_{t_k}$), which ensures that the state at the start of each attempt cannot be a goal state.

- $\mathcal{G}_k$ denotes the set of goal states, $\mathcal{S}_k := \mathcal{S} \setminus \mathcal{G}_k$ the set of non-goal states and $\mathcal{A}$ the set of actions.
- The transition model $p_k$ is the same as the original $p$ except for the transitions exiting the goal states which are redirected as a self-loop, i.e., $p_k(s'|s,a) := p(s'|s,a)$ and $p_k(g|g,a) := 1$ for any $(s, s', a, g) \in \mathcal{S}_k \times \mathcal{S} \times \mathcal{A} \times \mathcal{G}_k$.
- The cost function $c_k$ is defined as follows: for any $a \in \mathcal{A}$, any goal state $g \in \mathcal{G}_k$ is zero-cost ($c_k(g,a) := 0$), while the non-goal costs are unitary ($c_k(s,a) := 1$ for $s \in \mathcal{S}_k$).

From [10], Asm. 1 and the positive non-goal costs $c_k$ entail that solving $M_k$ is a well-posed SSP problem and that there exists an optimal policy that is *proper* (i.e., that eventually reaches one of the goal states with probability 1 when starting from any $s \in \mathcal{S}_k$). Crucially, the objective of collecting a sample from the under-sampled states $\mathcal{G}_k$ coincides with the SSP objective of minimizing the expected cumulative cost to reach a goal state in $M_k$.

**Step 2.** Since $p_k$ is unknown, we cannot directly compute the shortest-path policy for $M_k$. Instead, leveraging the samples collected so far, we apply an extended value iteration scheme for SSP which implicitly skews the empirical transitions $\widehat{p}_k$ towards reaching the goal states. This procedure can be done efficiently as shown in [50] (see App. A), and it outputs an *optimistic* shortest-path policy $\widetilde{\pi}_k$.

**Step 3.** $\widetilde{\pi}_k$ is then executed with the aim of quickly reaching an under-sampled state. Along its trajectory, the counter $N_t$ is updated for each visited state-action. Because of the error in estimating the model, $\widetilde{\pi}_k$ may never reach one of the goal states (i.e., it may not be proper in $p_k$). Thus $\widetilde{\pi}_k$ is executed until either one of the goals in $\mathcal{G}_k$ is reached, or the number of visits is doubled in a state-action pair in $\mathcal{S}_k \times \mathcal{A}$, a standard termination condition first introduced in [28]. If a sought-after goal state is reached, the agent executes an under-sampled action according to the current sampling requirements at that state. At the end of each attempt, the statistics (e.g., model estimate) are updated.

The algorithmic design of GOSPRL is conceptually simple and can flexibly incorporate various modifications driven by slightly different objectives or prior knowledge, without altering Thm. 1 (cf. App. B).

## 3.2 Sample Complexity Guarantee of GOSPRL

Thm. 1 establishes the sample complexity guarantee of GOSPRL (Alg. 1).

**Theorem 1.** *Under Asm. 1 and 2, for any sampling requirement sequence $b = (b_t)_{t \geq 1}$ and any confidence level $\delta \in (0,1)$, the sample complexity of GOSPRL is bounded as*

$$\mathcal{C}(\mathrm{GOSPRL}, b, \delta) = \widetilde{O}\Big(\overline{B}D + D^{3/2}S^2A\Big), \tag{1}$$

$$\mathcal{C}(\mathrm{GOSPRL}, b, \delta) = \widetilde{O}\Big(\sum_{s \in \mathcal{S}} \big(D_s \overline{b}(s) + D_s^{3/2}S^2A\big)\Big), \tag{2}$$

*where the $\widetilde{O}$ notation hides logarithmic dependencies on $S$, $A$, $D$, $1/\delta$ and $\overline{b}(s) := \sum_{a \in \mathcal{A}} \overline{b}(s,a)$ and $\overline{B} := \sum_{s \in \mathcal{S}} \overline{b}(s)$. Recall that $D_s \leq D$ is the SSP-diameter of state $s$ and captures the difficulty of collecting a sample at state $s$ starting at any other state in the MDP.*

We notice that in practice GOSPRL stops at the first *random* step $\tau$ at which the sampling requirement $b_\tau(s,a)$ is achieved for all $(s,a)$. Thm. 1 provides a worst-case upper bound on the stopping time of GOSPRL using the possibly loose bound $b_\tau(s,a) \leq \overline{b}(s,a)$. On the other hand, in the special case of $b : \mathcal{S} \to \mathbb{N}$ when the requirements are both time-independent (i.e., given as initial input to the algorithm) and action-independent, the actual sampling requirement $b(s)$ (resp. $B := \sum_{s \in \mathcal{S}} b(s)$) replaces $\overline{b}(s)$ (resp. $\overline{B}$) in the bound. In the following, we consider this case for the ease of exposition.

**Proof idea.** The key step (see App. C for the full derivation) is to link the sample complexity of GOSPRL to the regret accumulated over the sequence of multi-goal SSP problems $M_k$ generated across multiple attempts. Indeed we can define the regret at attempt $k$ as the gap between the performance of the SSP-optimal policy $\pi_k^\star$ solving $M_k$ (i.e., the minimum expected number of steps to reach any of the states in $\mathcal{G}_k$ starting from $s_{t_k}$) and the actual number of steps executed by GOSPRL before terminating the attempt. While the SSP regret minimization analysis of [45] assumes that the goal is fixed, we show that it is possible to bound the regret accumulated across different attempts for any arbitrary sequence of goals. The proof is concluded by bounding the cumulative performance of the SSP-optimal policies and it leads to the bound $\widetilde{O}(BD + D^{3/2}S^2A)$ where $B := \sum_{s \in \mathcal{S}} b(s)$. On the other hand, the refined bound in Eq. 2 requires a more careful analysis, where we no longer directly

translate regret bounds into sample complexity and we rather focus on relating the performance to state-dependent quantities $D_s$ and $b(s)$. Finally, we show that the extension to the general case of time-dependent action-dependent sampling requirements is straightforward and obtain Thm. 1.

**Interpretation of Thm. 1.** We can decompose Eq. 1 as a linear term in $B$ and a constant term. In the regime of large sample requirements (i.e., large $B$), the sample complexity thus reduces to $\widetilde{O}(BD)$, which adds at most an extra "cost" factor of $D$ w.r.t. an $\mathcal{SO}$. As this may be loose in many cases, the more refined analysis of Eq. 2 stipulates a cost of $D_s$ to collect a sample at state $s$, which better captures the connectivity of the MDP. In fact the lower bound in Lem. 1 shows that this cost of $D_s$ is *unavoidable in the worst case*, and that GOSPRL is only constant and logarithmic terms off w.r.t. to the best sample complexity that can be achieved in the worst case. While an extra attempt of refinement would be to avoid being worst-case w.r.t. the starting state in the definition of $D_s$,[4] this seems particularly challenging as the randomness of the environment makes it hard to control and analyze the sequence of states traversed by the agent. Also note that existing bounds in SSP [50, 45] are only worst-case and it remains an open question to derive finer (e.g., problem-dependent) bounds in SSP and how they could be leveraged in our case.

**Optimal solution.** GOSPRL targets a *greedy-optimal* strategy, which seeks to sequentially minimize each time to reach an under-sampled state. Alternatively, one may wonder if it is possible to design a learning algorithm that approaches the performance of the *exact-optimal* solution, i.e., a (non-stationary) policy explicitly minimizing the number of steps required to fulfill the sampling requirements.[5] Such strategy can be characterized as the optimal policy of an SSP problem for an MDP with state space augmented by the current sampling requirements and goal state corresponding to the case when all desired samples are collected. Even under known dynamics, the computational complexity of computing the optimal policy in this MDP (e.g., via value iteration) is exponential (scaling in $B^S$). When the dynamics is unknown, it appears highly challenging to obtain any learning algorithm whose performance is comparable to the exact-optimal strategy for any finite sample requirement $B$.

**Beyond Communicating MDPs.** In App. E we design an extension of GOSPRL to poorly or weakly communicating environments. In this setting, it is expected to assess online the "feasibility" of certain sampling requirements and discard them whenever associated to states that are *too difficult* to reach or unreachable. Given as input a "reachability" threshold $L$, we derive sample complexity guarantees for our variant of GOSPRL where the (possibly large or infinite) diameter $D$ is fittingly replaced by $L$.

# 4 Applications of GOSPRL

An appealing feature of GOSPRL is that it can be integrated with techniques that compute the (fixed or adaptive) sampling requirements to readily obtain an online RL algorithm with theoretical guarantees. In this section we focus on three specific problems where in our decoupled approach the $\mathcal{SO}$-based algorithm is either trivial or can be directly extracted from existing literature, and its combination with the sample collection strategy of GOSPRL yields improved or novel guarantees. Other applications (e.g., PAC-policy learning, diameter estimation, bridging bandits and MDPs) are illustrated in App. I.

## 4.1 Sparse Reward Discovery (TREASURE)

A number of recent methods focus on the state-space coverage problem, where each state in the MDP needs to be reached as quickly as possible. This problem is often motivated by environments where a one-hot reward signal, called the *treasure*, is hidden and can only be discovered by reaching a specific state and taking a specific action. Not only the environment but also the treasure state-action pair is unknown, and the agent does not receive any side information to guide its search (e.g., a measure of closeness to the treasure). Thus the agent must perform exhaustive exploration to find the treasure.

**Definition 2.** *Given a confidence $\delta \in (0, 1)$, the TREASURE sample complexity of a learning algorithm $\mathfrak{A}$ is defined as $\mathcal{C}_{\text{TREASURE}}(\mathfrak{A}, \delta) := \min\{t > 0 : \mathbb{P}(\forall(s, a) \in \mathcal{S} \times \mathcal{A}, N_t(s, a) \geq 1) \geq 1 - \delta\}.$*

---

[4]For instance, consider a simple deterministic chain with a requirement of one sample per state. If the agent starts on the leftmost state, then a policy that keeps moving right has sample complexity $S$ without extra factor $D$.

[5]Notice that as illustrated in the lower bound of Lem. 1, the exact-optimal and greedy-optimal have the same performance in the worst case.

In this case, a $\mathcal{SO}$-based algorithm would immediately solve the problem by collecting one sample from each state-action pair. As a result, we can directly apply GOSPRL for TREASURE by simply setting $b(s,a) = 1$ for each $(s,a)$ and from Thm. 1 with $B = SA$ we obtain the following guarantee.

**Lemma 2.** GOSPRL *with* $b(s,a) = 1$ *verifies* $\mathcal{C}_{\mathrm{TREASURE}}(\mathrm{GOSPRL}, \delta) = \widetilde{O}\big(D^{3/2}S^2A\big)$.

We now compare this result to alternative approaches to the problem, showing that GOSPRL has state-of-the-art guarantee for TREASURE (see App. G for details).

- First, reward-free methods such as [30, 64, 31, 39] are designed for finite-horizon problems so their guarantees cannot be directly translated to sample complexity for the TREASURE problem. Nonetheless, we draw inspiration from their algorithmic principles and analyze a *reward-free* variant of UCRL2 [28, 22]. Specifically we consider 0/1-UCRL, which runs UCRL by setting a reward of 1 to under-sampled states and 0 otherwise. However, we obtain a TREASURE sample complexity for 0/1-UCRL of $\widetilde{O}\big(\sum_{s \in \mathcal{S}} D_s^3 S^2 A\big)$, which is always worse than the bound in Lem. 2.
- Second, we can adapt the MAXENT approach [26] to state-action coverage so that it targets a policy whose stationary state-action distribution $\lambda$ maximizes $H(\lambda) := -\sum_{s,a} \lambda(s,a) \log \lambda(s,a)$. While optimizing this entropy does not provably solve TREASURE, it encourages us to take a "worst-case" approach w.r.t. the state-action visitations, and rather maximize $F(\lambda) := \min_{(s,a) \in \mathcal{S} \times \mathcal{A}} \lambda(s,a)$. We show that the learning algorithm of [17] instantiated to maximize $F$ yields a TREASURE sample complexity of at least $\Omega\big(\min\{D^2S^2A/(\omega^\star)^2, D^3/(\omega^\star)^3\}\big)$ with $\omega^\star := \min_\lambda F(\lambda) \leq (SA)^{-1}$, which is significantly poorer than Lem. 2. In fact, in contrast to MAXENT-inspired methods that optimize for a single *stationary* policy, GOSPRL realizes a non-stationary strategy that gradually collects the required samples by tackling successive learning problems.

## 4.2 Model Estimation (MODEST)

We now study the problem of accurately estimating the unknown transition dynamics in a reward-free communicating environment. The objective was recently introduced in [54] and we refer to it as the *model-estimation problem*, or MODEST for short.

**Definition 3.** *Given an accuracy level* $\eta > 0$ *and a confidence level* $\delta \in (0,1)$, *the* MODEST *sample complexity of an online learning algorithm* $\mathfrak{A}$ *is defined as*

$$\mathcal{C}_{\mathrm{MODEST}}(\mathfrak{A}, \eta, \delta) := \min\big\{t > 0 : \mathbb{P}\big(\forall(s,a) \in \mathcal{S} \times \mathcal{A}, \|\widehat{p}_{\mathfrak{A},t}(\cdot|s,a) - p(\cdot|s,a)\|_1 \leq \eta\big) \geq 1 - \delta\big\},$$

*where* $\widehat{p}_{\mathfrak{A},t}$ *is the estimate (i.e., empirical average) of the transition dynamics* $p$ *after* $t$ *time steps.*

Unlike in TREASURE, here the sampling requirements are not immediately prescribed by the problem. To define the $\mathcal{SO}$-based algorithm we first upper-bound the estimation error using an empirical Bernstein inequality and then invert it to derive the amount of samples $b_t(s,a)$ needed to achieve the desired level of accuracy $\eta$ (see App. F). Specifically, letting $\widehat{\sigma}_t^2(s'|s,a) := \widehat{p}_t(s'|s,a)(1 - \widehat{p}_t(s'|s,a))$ be the estimated variance of the transition from $(s,a)$ to $s'$ after $t$ steps, we set

$$b_t(s,a) := \left\lceil \frac{57(\sum_{s'} \widehat{\sigma}_t(s'|s,a))^2}{\eta^2} \log^2\left(\frac{8e(\sum_{s'} \widehat{\sigma}_t(s'|s,a))^2\sqrt{2SA}}{\sqrt{\delta}\eta}\right) + \frac{24S}{\eta}\log\left(\frac{24S^2A}{\delta\eta}\right)\right\rceil. \quad (3)$$

Since the estimated variance changes depending on the samples observed so far, the sampling requirements are *adapted* over time. Given that $\widehat{\sigma}_t^2(s'|s,a) \leq 1/4$, $b_t(s,a)$ is always bounded so Thm. 1 provides the following guarantee.

**Lemma 3.** *Let* $\Gamma := \max_{s,a}\|p(\cdot|s,a)\|_0 \leq S$ *be the maximal support of* $p(\cdot|s,a)$ *over the state-action pairs* $(s,a)$. *Running* GOSPRL *with the sampling requirements in Eq. 3 yields*

$$\mathcal{C}_{\mathrm{MODEST}}(\mathrm{GOSPRL}, \eta, \delta) = \widetilde{O}\Big(\frac{D\Gamma SA}{\eta^2} + \frac{DS^2A}{\eta} + D^{3/2}S^2A\Big).$$

Lem. 3 improves over the result of [54] in two important aspects. First, the latter suffers from an inverse dependency on the stationary state-action distribution that optimizes a proxy objective function used in the derivation of their algorithm. Second, while [54] requires an ergodicity assumption, Lem. 3 is the first sample complexity result for MODEST in the more general communicating setting.

### 4.3 Goal-Free & Cost-Free Exploration in Communicating MDPs

We finally delve into the paradigm of *reward-free exploration* introduced by [30]: the objective of the agent is to collect enough information during the reward-free exploration phase, so that it can readily compute a near-optimal policy once *any* reward function is provided. The problem has been analyzed in the *finite-horizon* setting [e.g., 30, 39, 64]. Here we study the more general and challenging setting of *goal-conditioned* RL.[6] We define the *goal-free cost-free* objective as follows: after the exploration phase, the agent is expected to compute a near-optimal goal-conditioned policy for *any* goal state and *any* cost function (w.l.o.g. we consider a maximum possible cost $c_{\max} = 1$). Recall that given a goal state $g$ and costs $c$, the (possibly unbounded) value function of a policy $\pi$ is

$$V^\pi(s \to g) := \mathbb{E}\left[ \sum_{t=1}^{\tau_\pi(s \to g)} c(s_t, \pi(s_t)) \mid s_1 = s \right].$$

Given a slack parameter $\theta \in [1, +\infty]$, we say that a policy $\widehat{\pi}$ is $(\varepsilon, \theta)$-optimal if [7]

$$V^{\widehat{\pi}}(s \to g) \leq \min_{\pi: \mathbb{E}[\tau_\pi(s \to g)] \leq \theta D_{s,g}} V^\pi(s \to g) + \varepsilon.$$

In this setting, constructing an efficient $\mathcal{SO}$-based algorithm is considerably more complex than TREASURE and MODEST. Relying on a sample complexity analysis for the fixed-goal SSP problem with a *generative model* [53], we define the (adaptive) number of samples needed in each state-action pair for our online objective. Although the number depends on the unknown diameter, we estimate $D$ using GOSPRL. The resulting sequence of sampling requirements is then fed online to GOSPRL. Combining the result of [53] and the properties of GOSPRL yields the following bound (see App. H).

**Lemma 4.** *Consider any MDP satisfying Asm. 1 and the goal-free cost-free exploration problem with accuracy level $0 < \varepsilon \leq 1$, confidence level $\delta \in (0, 1)$, minimum cost $c_{\min} \in [0, 1]$, slack parameter $\theta \in [1, +\infty]$. We can instantiate GOSPRL so that its exploration phase (i.e., number of time steps) is bounded with probability at least $1 - \delta$ by*

$$\widetilde{O}\left( \frac{D^4 \Gamma S A}{\omega \varepsilon^2} + \frac{D^3 S^2 A}{\omega \varepsilon} + \frac{D^3 \Gamma S A}{\omega^2} \right),$$

*where $\omega := \max\left\{ c_{\min}, \varepsilon/(\theta D) \right\} > 0$ (thus, either $c_{\min} = 0$ or $\theta = +\infty$, but not both simultaneously). Following the exploration phase, the algorithm can compute in the planning phase, for any goal state $g \in \mathcal{S}$ and any cost function $c$ in $[c_{\min}, 1]$, a policy $\widehat{\pi}_{g,c}$ that is $(\varepsilon, \theta)$-optimal.*

Lem. 4 establishes the first sample complexity guarantee for general goal-free, cost-free exploration. While the objective is demanding and the upper bound on the length of the exploration phase can be large, the main purpose of this result is to showcase how GOSPRL can be readily instantiated to tackle a challenging exploration problem for which no existing solution can be easily leveraged. Comparing our analysis to the finite-horizon objective of [30] reveals two interesting properties:

- **The goal-free aspect:** moving from finite-horizon to goal-conditioned renders *unavoidable* both the communicating requirement (Asm. 1) and the bound's dependency on the unknown diameter $D$ (which partly captures the role of the known horizon $H$ in the bound of [30]).
- **The cost-free aspect:** in contrast to finite-horizon, the value of $c_{\min}$ has an important impact on the type of performance guarantees we can obtain; in particular our analysis distinguishes between positive and non-negative costs (as also done in existing SSP analysis [11, 50, 45]).

## 5 Experiments

In this section we report a preliminary numerical validation of our theoretical findings. While GOSPRL can be integrated in many different contexts, here we focus on the problems where our theory suggests that GOSPRL performs better than state-of-the-art online learning methods.

---

[6]While an approach was proposed in [52], it is restricted to considering only the *incrementally* attainable goal states from a resettable reference state $s_0$.

[7]This reduces to standard $\varepsilon$-optimality for $\theta = +\infty$. We only consider $\theta < +\infty$ in the case of minimum possible cost $c_{\min} = 0$ and it ensures that the algorithm targets proper policies (see App. H).

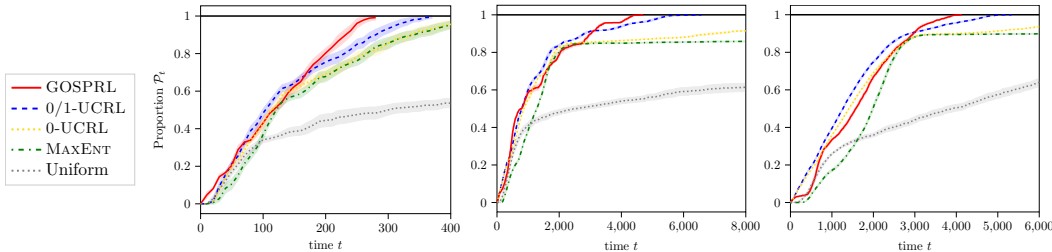

Figure 1: TREASURE-10 problem (i.e., with $b(s, a) = 10$): Proportion $\mathcal{P}_t$ of states meeting the requirements at time $t$, averaged over 30 runs. By definition of the sample complexity, the metric of interest is *not* the rate of increase of $\mathcal{P}_t$ over time but only the time needed to reach the line of success $\mathcal{P}_t = 1$. *Left:* 6-state RiverSwim, *Center:* 24-state corridor gridworld, *Right:* 43-state 4-room gridworld (see App. K for details on the domains).

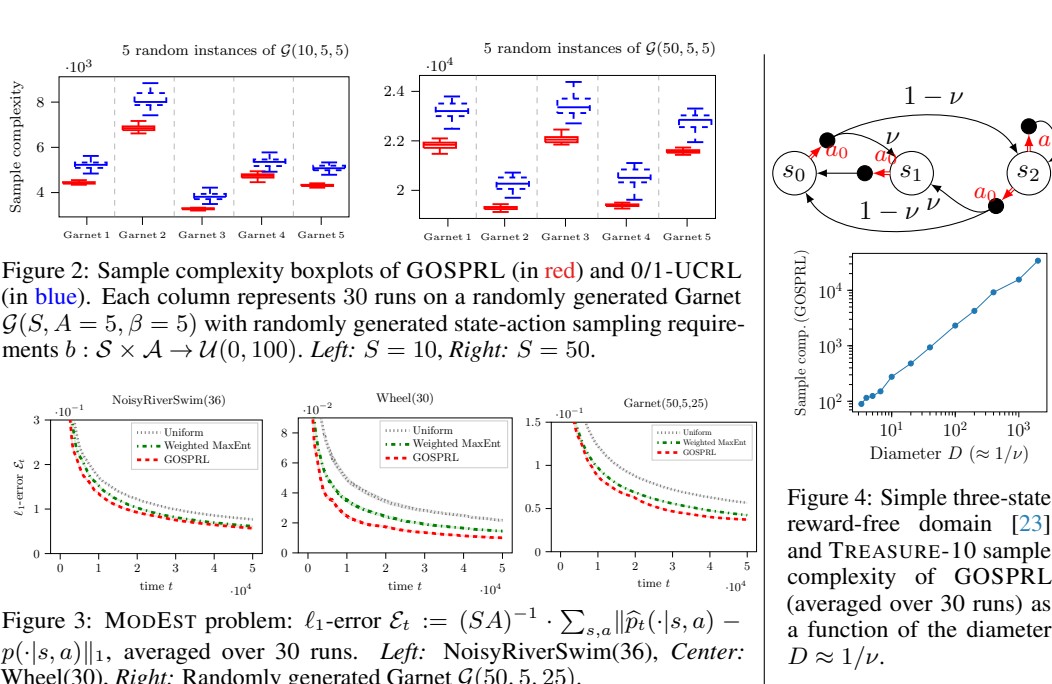

Figure 2: Sample complexity boxplots of GOSPRL (in red) and 0/1-UCRL (in blue). Each column represents 30 runs on a randomly generated Garnet $\mathcal{G}(S, A = 5, \beta = 5)$ with randomly generated state-action sampling requirements $b : \mathcal{S} \times \mathcal{A} \to \mathcal{U}(0, 100)$. *Left:* $S = 10$, *Right:* $S = 50$.

Figure 3: MODEST problem: $\ell_1$-error $\mathcal{E}_t := (SA)^{-1} \cdot \sum_{s,a} \|\widehat{p}_t(\cdot|s, a) - p(\cdot|s, a)\|_1$, averaged over 30 runs. *Left:* NoisyRiverSwim(36), *Center:* Wheel(30), *Right:* Randomly generated Garnet $\mathcal{G}(50, 5, 25)$.

Figure 4: Simple three-state reward-free domain [23] and TREASURE-10 sample complexity of GOSPRL (averaged over 30 runs) as a function of the diameter $D \approx 1/\nu$.

**TREASURE-type problem.** We consider a TREASURE-type problem (Sect. 4.1), where for all $(s, a)$ we set $b(s, a) = 10$ instead of 1 (we call it the TREASURE-10 problem).[8] We begin by showing in Fig. 4 that it is easy to construct a worst-case problem where the sample complexity scales linearly with the diameter, which is consistent with the theoretical discussion in Sect. 2 and 3.

We compare to two heuristics based on UCRL2B [28, 22]: 0-UCRL, where the reward used in computing the optimistic policy is set proportional to $([N(s, a) - b(s, a)]^+)^{-1/2}$, and 0/1-UCRL with reward 1 for undersampled state-action pairs and 0 otherwise. We also compare with the MAXENT algorithm [17] that maximizes entropy over the state-action space, and with a uniformly random baseline policy. We test on the RiverSwim domain [48] and various gridworlds (see App. K for details and more results). Fig. 1 reports the proportion $\mathcal{P}_t$ of states that satisfy the sampling requirements at time $t$. Our metric of interest is the time needed to collect all required samples, and we see that GOSPRL reaches the $\mathcal{P}_t = 1$ line of success consistently, and faster than 0/1-UCRL, while the other heuristics struggle. The steady increase of $\mathcal{P}_t$ illustrates GOSPRL's design to progressively meet the sampling requirements, and not exhaust them state after state.

---

[8]Since GOSPRL and our baselines are all based on upper confidence bounds, they tend to display similar behaviors in the initial phases of learning, since the estimates when $N(s, a) = 0$ are similar. As the number of samples required in each state-action increases, the difference between the algorithms' design starts making a real difference in the behavior and eventually their performance. This is why we study here TREASURE-10 instead of the TREASURE-1 problem for which empirical performance is comparable between learning algorithms.

**Random MDPs and sampling requirements.** To study the generality of GOSPRL to collect arbitrary sought-after samples, we further compare GOSPRL with 0/1-UCRL which is the best heuristic from the previous experiment. We test on a variety of randomly generated *configurations*, that we define as follows: each configuration corresponds to **i)** a randomly generated Garnet environment $\mathcal{G}(S, A, \beta)$ (with $S$ states, $A$ actions and branching factor $\beta$, see [12]), and **ii)** randomly generated requirements $b(s, a) \in \mathcal{U}(0, \overline{U})$, where the maximum budget is set to $\overline{U} = 100$ to have a wide range of possible requirements across each environment. The boxplots in Fig. 2 provide aggregated statistics on the sample complexity for different configurations. We observe that GOSPRL consistently meets the sampling requirements faster than 0/1-UCRL, as well as suffers from lower variance across runs.

**MODEST problem.** Finally, we empirically evaluate GOSPRL for the MODEST problem (Sect. 4.2). We compare to the fully online WEIGHTEDMAXENT heuristic, which weighs the state-action entropy components with an optimistic estimate of the next-state transition variance and was shown in [54] to perform empirically better than algorithms with theoretical guarantees. We test on the two environments (NoisyRiverSwim and Wheel) proposed in [54] for their high level of stochasticity, as well as on a randomly generated Garnet. To facilitate the comparison, we consider a GOSPRL-for-MODEST algorithm where the sampling requirements are computed using a decreasing error $\eta$ (see App. K for details). We observe in Fig. 3 that GOSPRL outperforms the WEIGHTEDMAXENT heuristic.

# 6 Conclusion

In this paper, we introduced the online learning problem of simulating a sampling oracle (Sect. 2) and derived the algorithm GOSPRL with its sample complexity guarantee (Sect. 3). We then illustrated how it can be used to tackle in a unifying fashion a variety of applications without having to design a specific online algorithm for each, while at the same time obtaining improved or novel sample complexity guarantees (Sect. 4). Going forward, we believe that GOSPRL can be used as a competitive off-the-shelf baseline when a new application is introduced.

An exciting direction of future investigation is to extend the general sample collection problem and its various applications beyond the tabular setting. Handling a continuous state space or linear function approximation requires redefining the notion of reaching a specific state (e.g., via adequate discretization or by considering requirements based on the covariance matrix). Studying the SSP problem beyond tabular may provide insights, as recently initiated in [56] in linear function approximation under the assumption that all policies are proper. On the more algorithmic side, GOSPRL hinges on knowing the sampling requirement function $b_t$ and deriving a shortest-path policy $\widetilde{\pi}$. Interestingly, we can identify algorithmic counterparts to both modules in deep RL. The computation of $\widetilde{\pi}$ can be entrusted to a goal-conditioned network (using e.g., [2]), while the specification of $b_t$ can be related to goal-sampling selection mechanisms that elect hard-to-reach [21] or rare [42] states as goals.

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
