# Appendix

## Table of Contents

## A  Efficient Computation of Optimistic SSP Policy

In this section, we recall how to compute an optimistic stochastic shortest path (SSP) policy using an extended value iteration (EVI) scheme tailored to SSP, as explained in [50]. Here we leverage a Bernstein-based construction of confidence intervals, as done by e.g., [22, 45]. For details on the SSP formulation, we refer to e.g., [9, Sect. 3].

Consider as input an SSP-MDP instance $M^\dagger := \langle \mathcal{S}^\dagger, \mathcal{A}, c, p, s^\dagger \rangle$, with goal $s^\dagger$, non-goal states $\mathcal{S}^\dagger = \mathcal{S} \setminus \{s^\dagger\}$, actions $\mathcal{A}$, unknown dynamics $p$, and known cost function with costs in $[c_{\min}, 1]$ where $c_{\min} > 0$. We assume that there exists at least one proper policy (i.e., that reaches the goal $s^\dagger$ with probability one when starting from any state in $\mathcal{S}^\dagger$). Note that in particular such condition is verified under Asm. 1. We denote by $N(s, a)$ the current number of samples available at the state-action pair $(s, a)$ and set $N^+(s, a) := \max\{1, N(s, a)\}$. We also denote by $\widehat{p}$ the current empirical average of transitions: $\widehat{p}(s'|s, a) = N(s, a, s')/N(s, a)$.

The algorithm first computes a set of plausible SSP-MDPs defined as

$$\mathcal{M}^\dagger := \{\langle \mathcal{S}^\dagger, \mathcal{A}, c, \widetilde{p}, s^\dagger \rangle \mid \widetilde{p}(s^\dagger|s^\dagger, a) = 1, \ \widetilde{p}(s'|s, a) \in \mathcal{B}(s, a, s'), \ \sum_{s'} \widetilde{p}(s'|s, a) = 1\},$$

where for any $(s, a) \in \mathcal{S}^\dagger \times \mathcal{A}$, $\mathcal{B}(s, a, s')$ is a high-probability confidence set on the dynamics of the true SSP-MDP $M^\dagger$. Specifically, we define the compact sets $\mathcal{B}(s, a, s') := [\widehat{p}(s'|s, a) - \beta(s, a, s'), \widehat{p}(s'|s, a) + \beta(s, a, s')] \cap [0, 1]$, where

$$\beta(s, a, s') := 2\sqrt{\frac{\widehat{\sigma}^2(s'|s, a)}{N^+(s, a)} \log\left(\frac{2SAN^+(s, a)}{\delta}\right)} + \frac{6 \log\left(\frac{2SAN^+(s, a)}{\delta}\right)}{N^+(s, a)},$$

where $\widehat{\sigma}^2(s'|s, a) := \widehat{p}(s'|s, a)(1 - \widehat{p}(s'|s, a))$ is the variance of the empirical transition $\widehat{p}(s'|s, a)$. Importantly, the choice of $\beta(s, a, s')$ guarantees that $M^\dagger \in \mathcal{M}^\dagger$ with high probability. Indeed, let us

now spell out the high-probability event. Denote by $\mathcal{E}$ the event under which for any time step $t \geq 1$ and for any state-action pair $(s, a) \in \mathcal{S} \times \mathcal{A}$ and next state $s' \in \mathcal{S}$, it holds that

$$|\widehat{p}_t(s'|s, a) - p(s'|s, a)| \leq \beta_t(s, a, s'). \tag{4}$$

Given the way the confidence intervals are constructed using the empirical Bernstein inequality [see e.g., 22, 45], we have $\mathbb{P}(\mathcal{E}) \geq 1 - \delta$. Throughout the remainder of the analysis, we will assume that the event $\mathcal{E}$ holds.

Once $\mathcal{M}^\dagger$ has been computed, the algorithm applies an extended value iteration (EVI) scheme to compute a policy with lowest optimistic value. Formally, it defines the extended optimal Bellman operator $\widetilde{\mathcal{L}}$ such that for any vector $\widetilde{v} \in \mathbb{R}^{S^\dagger}$ and non-goal state $s \in \mathcal{S}^\dagger$,

$$\widetilde{\mathcal{L}}\widetilde{v}(s) := \min_{a \in \mathcal{A}} \left\{ c(s, a) + \min_{\widetilde{p} \in \mathcal{B}(s,a)} \sum_{s' \in \mathcal{S}^\dagger} \widetilde{p}(s'|s, a)\widetilde{v}(s') \right\}.$$

We consider an initial vector $\widetilde{v}_0 := 0$ and set iteratively $\widetilde{v}_{i+1} := \widetilde{\mathcal{L}}\widetilde{v}_i$. For a predefined VI precision $\mu_{\text{VI}} > 0$, the stopping condition is reached for the first iteration $j$ such that $\|\widetilde{v}_{j+1} - \widetilde{v}_j\|_\infty \leq \mu_{\text{VI}}$. The policy $\widetilde{\pi}$ is then selected to be the optimistic greedy policy w.r.t. the vector $\widetilde{v}_j$. While $\widetilde{v}_j$ is *not* the value function of $\widetilde{\pi}$ in the optimistic model $\widetilde{p}$, which we denote by $\widetilde{V}^{\widetilde{\pi}}$, both quantities can be related according to the following lemma, which is a simple adaptation of [50, Lem. 4 & App. E]. We denote by $V^\star$ (resp. $\widetilde{V}^\star$) the optimal value function in the true (resp. optimistic) SSP instance.

**Lemma 5.** *Under the event $\mathcal{E}$, the following component-wise inequalities hold: 1)* $\widetilde{v}_j \leq V^\star$, *2)* $\widetilde{v}_j \leq \widetilde{V}^\star \leq \widetilde{V}^{\widetilde{\pi}}$, *3) If the* VI *precision level verifies* $\mu_{\text{VI}} \leq \frac{c_{\min}}{2}$, *then* $\widetilde{V}^{\widetilde{\pi}} \leq \left(1 + \frac{2\mu_{\text{VI}}}{c_{\min}}\right)\widetilde{v}_j$.

Note that for the purposes of GOSPRL (Alg. 1), the VI precision $\mu_{\text{VI}}$ can for example be selected as in [50] equal to $1/(2t_k)$ with $t_k$ the current time step, which only translates in a negligible, lower-order error in the sample complexity result of Thm. 1.

## B  Algorithmic Variants of GOSPRL

The algorithmic design of GOSPRL (Alg. 1) is conceptually simple and it can flexibly incorporate a number of modifications driven by the agent's desiderata or possible prior knowledge.

- Any non-unit SSP costs can be designed as long as they are positive and bounded: detering costs may e.g., be assigned to "trap" states with large negative environmental reward that the agent may seek to avoid.
- Penalizing the visitation of sufficiently visited states (with carefully selected larger-than-one costs) may give the agent incentive to "even out" its sample collection and thus avoid over-sampling some areas of the state-action space.
- It is possible to change the construction of the SSP problem and focus on specific goal states instead of considering all under-sampled states as goals. In practice, using such a *meta-goal* makes the optimal SSP policy more robust to noise. While the SSP solution to $M_k$ indeed seeks to reach the closest under-sampled state, random transitions may move the agent closer to any other state in $\mathcal{G}_k$ and this would naturally trigger the policy to focus on such closer state. On the other hand, providing the SSP policy with a single goal state may lead to much longer and wasteful attempts.
- Finally we remark that if the entire state space is initially under-sampled, any action would produce a "useful" sample and different heuristics can be implemented in prioritizing actions accordingly.

In the following, we delve into such goal-selection (App. B.1) and cost-shaping (App. B.2) variants of GOSPRL, which do not affect the sample complexity bound of Thm. 1.

### B.1  Selecting the goal state

In Alg. 1, each attempt $k$ casts as goals the states that are under-sampled w.r.t. the sampling requirements so far. As mentioned above, having such multiple goals is algorithmically appealing as it reduces the number of attempts that fail to collect a desired sample. Although specifically eliciting a single valid (i.e., under-sampled) goal state at each attempt may yield poorer performance, the

resulting sample complexity guarantee would be the same as in Thm. 1. We can then distinguish between two strategies for goal state selection.

The first strategy prioritizes the states that appear harder to be successfully sampled. This is sensible when the aim is to have the most even possible sample collection over time so as to shy away from purely local exploration. For instance, the learning agent can select as goal state the least-sampled state so far, i.e., $\overline{s}_k \in \arg\min_{s \in \mathcal{G}_k} N_k(s)$.

On the other hand, the second strategy prioritizes the states that appear easier to be successfully sampled. This is sensible when the objective is to solely meet the total sampling requirements as fast as possible. For instance, the agent can select the state with the best current ratio "successful sampling" / "attempted sampling", in order to encourage the algorithm to exploit areas of the state space that it supposedly masters well, i.e.,

$$\overline{s}_k \in \arg\min_{s \in \mathcal{G}_k} \frac{\#\{i \in [k-1] : \overline{s}_i = s \text{ and } N_{i+1}(s) = N_i(s) + 1\}}{\#\{i \in [k-1] : \overline{s}_i = s\}}.$$

We point out that the possibility of not considering all undersampled states as goal states may be particularly relevant during the *initial phase* of GOSPRL, which corresponds to the time steps when *all* the states are under-sampled and thus are goal states $\mathcal{G}_k$. This initial phase may furthermore be quite long when the sampling requirements verify $b(s) \gg 1$ for all $s \in \mathcal{S}$ (e.g., in the MODEST problem of Sect. 4.2). Naturally, the execution of any policy in the initial phase will collect "relevant" samples, until we get $\mathcal{G}_k \subsetneq \mathcal{S}$. As such, the sample complexity guarantee of Thm. 1 is the same whatever the strategy employed during the initial phase. In our experiments (Sect. 5 and App. K), we consider an initial phase where the goal states $s$ are selected as those minimizing the "remaining budget" $b(s) - N(s)$ in the case of state-only requirements, or $\sum_a \max\{b(s, a) - N(s, a), 0\}$ for state-action requirements. This has the effect of shortening the length of the initial phase.

## B.2 Cost-shaping the trajectories

Instead of considering unit costs, it is possible to introduce varying costs for the SSP instance considered at each attempt. Indeed, if we seek to penalize the state-action pair $(s, a)$ at an attempt $k$, we can simply set the cost $c_k(s, a)$ to a quantity larger than 1. Imposing the costs to belong to the interval $[1, \overline{c}]$, where $\overline{c} \geq 1$ is a constant that upper bounds all possible costs, the resulting sample complexity bound in Thm. 1 stays the same as it only inherits a constant multiplicative factor of $\overline{c}$.

First, this cost-sensitive procedure implies that if the agent has a prior knowledge or requirement that some (resp. actions) should be avoided, the agent can straightforwardly set the maximal cost $\overline{c}$ to such states (resp. actions) in order to discourage their visitation (resp. their execution). We show this behavior in a simple experiment in App. K.

Second, while GOSPRL is attentive in avoiding under-sampling (i.e., to achieve a desired threshold of state visitations), it is not mindful in avoiding over-sampling certain state-action pairs. Some recent approaches (e.g., [26, 17, 18]) perform a sort of "distribution tracking" (via the Frank-Wolfe algorithm), achieving a more "stable" and "smooth" behavior which attempts to limit *both* over-sampling and under-sampling. Unfortunately, their direct application struggles to provably enforce a minimum amount of sampling, as we explain in Sect. 4.1 and App. G. Yet we can draw inspiration from these techniques to give the agent incentive to "even out" the sample collection w.r.t. the requirements $b(s)$. A way to mitigate this effect is to encourage the agent to visit each state $s$ with empirical frequency close to the target frequency $b(s)/B$. To do so, we can propose to penalize the visitation of sufficiently visited states by considering cost-sensitive SSP instances that verify the following informal claim.

**Claim 1.** *In order to even out the sample collection w.r.t. the final requirements, at each attempt $k$, each cost $c_k(s)$ should scale as $\phi(N_k(s))$, where $N_k(s)$ is the number of samples collected so far at state $s$, and $\phi$ is a non-decreasing function which is either clipped or re-scaled in the interval $[1, \overline{c}]$.*

This idea is fairly intuitive and, although it seems complicated to quantify the extent to which the sample collection would be effectively evened out, we now provide a theoretically grounded justification behind Claim 1 which draws a parallel between reinforcement learning and convex optimization (namely, the Frank-Wolfe algorithm).

On the one hand, for a given starting state $s_0$, goal state $\overline{s}$ and costs $c$, the SSP problem can be solved with linear programming over the dual space, where the optimization variables $\lambda(s, a)$, known as

occupation measures, represent the expected number of times action $a$ is executed in state $s$. The program can be written as follows (see e.g., [20, 55])

$$\min_{\lambda} \quad \sum_{s,a} c(s,a)\lambda(s,a)$$

subject to

(i) $\lambda(s,a) \geq 0 \quad \forall (s,a) \in \mathcal{S} \times \mathcal{A}$,     (iv) $\mu_{\text{out}}(s_0) - \mu_{\text{in}}(s_0) = 1$,

(ii) $\mu_{\text{in}}(s) = \sum_{s',a} \lambda(s',a)p(s|s',a) \quad \forall s \in \mathcal{S}$,     (v) $\mu_{\text{out}}(s) = \sum_{a} \lambda(s,a) \quad \forall s \in \mathcal{S} \setminus \{\bar{s}\}$,

(iii) $\mu_{\text{out}}(s) - \mu_{\text{in}}(s) = 0 \quad \forall s \in \mathcal{S} \setminus \{s_0, \bar{s}\}$,     (vi) $\mu_{\text{in}}(\bar{s}) = 1$.

This dual formulation can be interpreted as a flow problem, where the constraints (ii) and (v) respectively define the expected flow entering and leaving state $s$; (iii) is the flow conservation principle; (iv) and (vi) define respectively the starting state and the goal state. The objective function captures the minimization of the total expected cost to reach the goal state from the starting state. Once the optimal solution $\lambda^\star$ is computed, the optimal policy is $\pi^\star(a|s) = \lambda^\star(s,a)/\mu^\star_{\text{out}}(s)$ and is guaranteed to be deterministic, i.e., for all $s$ such that $\mu^\star_{\text{out}}(s) > 0$, we have $\lambda^\star(s,a) > 0$ for exactly one action $a$.

On the other hand, we seek to "even out" the sample collection w.r.t. the requirements $b(s)$. A natural way to do so can be to encourage the agent to visit each state $s$ with empirical frequency close to the target frequency $\frac{b(s)}{B}$. For instance, two objective functions achieving this are the following

$$\min_{\lambda} \mathcal{L}_1(\lambda) := \frac{1}{2} \sum_{s \in \mathcal{S}} \left( \frac{b(s)}{B} - \sum_{a \in \mathcal{A}} \lambda(s,a) \right)^2 ; \qquad \min_{\lambda} \mathcal{L}_2(\lambda) := \sum_{s \in \mathcal{S}} \sum_{a \in \mathcal{A}} \lambda(s,a) \log \left( \frac{\sum_{a \in \mathcal{A}} \lambda(s,a)}{b(s)/B} \right).$$

The first objective function is studied in [18] as the "Space Exploration" problem, while the second KL-divergence objective is tackled in [26, 17]. Both methods leverage a Frank-Wolfe algorithmic design, since $\mathcal{L}_1$ and $\mathcal{L}_2$ are both convex and Lipschitz-continuous in $\lambda$. Following [26, 17, 18], for a given attempt $k$ with empirical state-action frequencies $\widetilde{\lambda}_k$, the occupation measure to be targeted should minimize the following inner product: $\min_{\lambda} \langle \nabla \mathcal{L}(\widetilde{\lambda}_k), \lambda \rangle$. The gradients of the two objective functions above are

$$\nabla \mathcal{L}_1(\lambda) = \phi_1 \left( \sum_{a \in \mathcal{A}} \lambda(\cdot, a) \right), \text{ with } \phi_1(x) = x - \frac{b(s)}{B};$$

$$\nabla \mathcal{L}_2(\lambda) = \phi_2 \left( \sum_{a \in \mathcal{A}} \lambda(\cdot, a) \right), \text{ with } \phi_2(x) = \log(x) + 1 - \log \left( \frac{b(s)}{B} \right).$$

Note that both $\phi_1$ and $\phi_2$ are non-decreasing functions. As a result, the $s$-th component of $\nabla \mathcal{L}(\widetilde{\lambda}_k)$ scales with $\sum_a \widetilde{\lambda}_k(s,a)$, i.e., with $N_k(s)$. Furthermore, in light of the contrained linear program above, the costs $c_k(s)$ should scale with the $s$-th component of $\nabla \mathcal{L}(\widetilde{\lambda}_k)$. This gives informal grounds to Claim 1 of having the SSP costs at each state grow with a non-decreasing function in the state visitations. It remains an open question whether it is possible to show if this approach yields a provable improvement in the sample complexity of GOSPRL, or quantify the extent to which it succeeds in evening out the sample collection w.r.t. the final requirements (i.e., by obtaining small values for the objective functions $\mathcal{L}_1$ or $\mathcal{L}_2$).

## C   Proof of Theorem 1

We first prove the special case where the sampling requirements are time- and action-independent, i.e., $b : \mathcal{S} \to \mathbb{N}$.

**Corollary 1.** *Under Asm. 1, for any input sampling requirements $b : \mathcal{S} \to \mathbb{N}$ with $B := \sum_{s \in \mathcal{S}} b(s)$ and for any confidence level $\delta \in (0,1)$,*

$$\mathcal{C}(\text{GOSPRL}, b, \delta) = \widetilde{O}\left( BD + D^{3/2}S^2 A \right), \tag{6}$$

$$\mathcal{C}(\text{GOSPRL}, b, \delta) = \widetilde{O}\left( \sum_{s \in \mathcal{S}} \left( D_s b(s) + D_s^{3/2} S^2 A \right) \right). \tag{7}$$

## C.1 Proof of Corollary 1

We denote by $\mathcal{E}$ the event under which the Bernstein inequalities stated in Eq. 4 hold simultaneously for each time step $t$ and each state-action-next-state triplet $(s, a, s') \in \mathcal{S} \times \mathcal{A} \times \mathcal{S}$, i.e.,

$$|\widehat{p}_t(s'|s,a) - p(s'|s,a)| \le \beta_t(s,a,s') := 2\sqrt{\frac{\widehat{\sigma}_t^2(s'|s,a)}{N_t^+(s,a)} \log\left(\frac{2SAN_t^+(s,a)}{\delta}\right)} + \frac{6\log\left(\frac{2SAN_t^+(s,a)}{\delta}\right)}{N_t^+(s,a)}.$$

Note that we have $\mathbb{P}(\mathcal{E}) \ge 1 - \delta$ from [22, 45].

We recall that at the beginning of each episode $j$, the under-sampled states $\mathcal{G}_j$ are cast as goal states.[9] GOSPRL then constructs an SSP-MDP instance $M_j := \langle \mathcal{S}_j, \mathcal{A}, p_j, c_j, \mathcal{G}_j \rangle$, where $\mathcal{G}_j$ encapsulates the goal states and $\mathcal{S}_j := \mathcal{S} \setminus \mathcal{G}_j$ the non-goal states. The transition model $p_j$ is the same as the original $p$ except for the transitions exiting the goal states which are redirected as a self-loop, i.e., $p_j(s'|s,a) := p(s'|s,a)$ and $p_j(g|g,a) := 1$ for any $(s, s', a, g) \in \mathcal{S}_j \times \mathcal{S} \times \mathcal{A} \times \mathcal{G}_j$. As for the cost function $c_j$, for any action $a \in \mathcal{A}$, any goal state $g \in \mathcal{G}_j$ is zero-cost (i.e., $c_j(g, a) := 0$), while the non-goal costs are unitary (i.e., $c_j(s, a) := 1$ for all $s \in \mathcal{S}_j$).

We now make more explicit the way the SSP optimistic policy is constructed at the beginning of any episode $j$. Denote by $\widehat{p}_j$ the empirical transitions of the induced SSP-MDP $M_j$. We consider the following confidence intervals $\beta_j'$ in the optimistic SSP policy computation from App. A

$$\forall (a, s') \in \mathcal{A} \times \mathcal{S}, \quad \forall s \notin \mathcal{G}_j, \ \beta_j'(s,a,s') := \beta_t(s,a,s'), \quad \forall s \in \mathcal{G}_j, \ \beta_j'(s,a,s') = 0.$$

We denote by $\widetilde{p}_j$ the optimistic model computed by the EVI scheme with such confidence intervals. Now, denoting by $\mathcal{P}(\mathcal{S})$ the power set of the state space $\mathcal{S}$, we have the following event inclusion

$$\begin{aligned} \mathcal{E} \subseteq \mathcal{E}' := \Big\{ &\forall j \ge 1, \ \forall \mathcal{G}_j \in \mathcal{P}(\mathcal{S}), \ \forall (a, s') \in \mathcal{A} \times \mathcal{S}, \\ &\forall s \notin \mathcal{G}_j, \ |\widetilde{p}_j(s'|s,a) - \widehat{p}_j(s'|s,a)| \le \beta_j'(s,a,s'), \\ &\forall s \in \mathcal{G}_j, \ \widetilde{p}_j(s|s,a) = 1 \Big\}. \end{aligned}$$

Indeed, the only transitions that are redirected from $p$ to $p_j$ are those that exit from states in $\mathcal{G}_j$ and they are set to deterministically self-loop, which implies that they do not contain any uncertainty. Note that $\mathcal{E}'$ is the event that we require to hold so that the SSP analysis goes through for *any* considered SSP-MDP $M_j$. From the inclusion above, we have that the event $\mathcal{E}'$ holds with probability at least $1 - \delta$, and we assume from now on that it holds.

We denote by $H_j$ the length of each episode $j$, specifically $H_j = \min_{h \ge 1}\{s_{j,h} \in \mathcal{G}_j\}$, where we denote by $s_{j,h}$ the $h$-th state visited during episode $j$. We denote by $\overline{s}_j := s_{j,H_j}$ the goal state in $\mathcal{G}_j$ that is reached at the end of episode $j$. Correspondingly, the starting state of each episode $j$, denoted by $\underline{s}_j$, also varies: if $j = 1$ it is the initial state $s_0$ of the learning interaction, otherwise it is equal to $\overline{s}_{j-1}$ which is the reached goal state at the end of the previous episode $j - 1$.[10] The important property is that both the starting state $\underline{s}_j$ and the goal states $\mathcal{G}_j$ are measurable (i.e., known and fixed) at the beginning of each episode $j$.

We define $R_J$ the regret after $J$ episodes as follows

$$R_J := \sum_{j=1}^{J} \sum_{h=1}^{H_j} c_j(s_{j,h}, a_{j,h}) - \sum_{j=1}^{J} \min_{\pi} V_j^{\pi}(\underline{s}_j), \tag{8}$$

---

[9]In the case of state-only requirements, a state $s$ is considered under-sampled if $\sum_{a \in \mathcal{A}} N_{t-1}(s, a) < b(s)$. In the case of state-action requirements, a state $s$ is considered under-sampled if $\exists a \in \mathcal{A}, N_{t-1}(s, a) < b(s, a)$.

[10]This choice of initial state for episodes is when we have state-only sampling requirements. If we instead have state-action requirements, the action taken at each reached goal state matters. In that case, when episode $j - 1$ reaches a goal state $\overline{s}_{j-1}$, the agent takes a relevant action $\overline{a}_{j-1}$ and we then consider that the starting state $\underline{s}_j$ at the next episode $j$ is distributed according to $p(\cdot|\overline{s}_{j-1}, \overline{a}_{j-1})$. The action $\overline{a}_{j-1}$ is naturally specified by the algorithm depending on the current and desired requirements $N(\overline{s}_{j-1}, \cdot)$ and $b(\overline{s}_{j-1}, \cdot)$, i.e., we should select $\overline{a}_{j-1} \in \{a \in \mathcal{A} : N(\overline{s}_{j-1}, a) < b(\overline{s}_{j-1}, a)\}$ with $N$ the state-action counter at the end of episode $j - 1$. We explain in App. K the way we select this action in our experiments.

where we denote by $V_j^\pi(s)$ the value function of a policy $\pi$ starting from state $s$ in the SSP-MDP instance $M_j$. We also denote by $\mathcal{C}(\text{GOSPRL}, b)$ the random variable of the total time accumulated by GOSPRL until the sampling requirements $b$ are met.

On the one hand, the regret $R_J$ can be lower bounded almost surely as follows

$$
\begin{aligned}
R_J &\overset{(a)}{=} \sum_{j=1}^J H_j - \sum_{j=1}^J \min_\pi \mathbb{E}\big[\tau_\pi(\underline{s}_j \to \mathcal{G}_j)\big] \\
&\overset{(b)}{=} \mathcal{C}(\text{GOSPRL}, b) - \sum_{j=1}^J \min_\pi \mathbb{E}\big[\tau_\pi(\underline{s}_j \to \mathcal{G}_j)\big] \\
&\overset{(c)}{\geq} \mathcal{C}(\text{GOSPRL}, b) - DB,
\end{aligned}
\tag{9}
$$

where (a) stems from the fact that all the non-goal costs are unitary, (b) comes from the definition of the index $J$ (i.e., the episode at which all the sampling requirements are met) and (c) combines that $J \leq B$ almost surely and that $\mathbb{E}\big[\tau_\pi(\underline{s}_j \to \mathcal{G}_j)\big] \leq D$ by definition of the diameter $D$.

On the other hand, retracing the analysis of [45], the derivation of the regret bound can be easily extended to varying initial states and varying (possibly multiple[11]) goal states across episodes, as long as they are all *known* to the learner at the beginning of each episode (which is our case here). In particular, the high-probability event is $\mathcal{E}' \supseteq \mathcal{E}$ defined above, which holds with probability at least $1 - \delta$. Under this event, we have from [45, Thm. 2.4] that GOSPRL satisfies

$$
R_J = \widetilde{O}\big(DS\sqrt{AJ} + D^{3/2}S^2 A\big).
\tag{10}
$$

Combining Eq. 9 and 10 yields that with probability at least $1 - \delta$, we have

$$
\mathcal{C}(\text{GOSPRL}, b) \leq \widetilde{O}\big(BD + DS\sqrt{AJ} + D^{3/2}S^2 A\big).
$$

Given that $J \leq B$ almost surely, we get

$$
\mathcal{C}(\text{GOSPRL}, b, \delta) \leq \widetilde{O}\big(BD + DS\sqrt{AB} + D^{3/2}S^2 A\big).
\tag{11}
$$

We now proceed with a separation of cases. If $B \geq S^2 A$, we have $DS\sqrt{AB} \leq BD$. Otherwise, if $B \leq S^2 A$, we have $DS\sqrt{AB} \leq D^{3/2}S^2 A$. This implies that the second summand in the $\widetilde{O}$ sum in Eq. 11 can be removed, which yields the first sought-after bound of Eq. 6.

In order to obtain the second more state-dependent bound of Eq. 7, the bound of Eq. 10 is too loose, hence we need to extend the analysis of [45] to bring out dependencies on $b(s)$ and $D_s$. In particular, we consider a similar decomposition in *epochs* and *intervals* that we carefully adapt for our purposes of varying goal states. The first epoch starts at the first time step and each epoch ends once the number of visits to some state-action pair is doubled. We denote by $\mathcal{G}_m$ the goal states that are considered during interval $m$ and by $D_{\mathcal{G}_m}$ the SSP-diameter of the goal states $\mathcal{G}_m$. The first interval starts at the initial time step and each interval $m$ (with goal states $\mathcal{G}_m$) ends once one of the four following conditions holds: (i) the length of the interval reaches $D_{\mathcal{G}_m}$; (ii) an unknown state-action pair is reached (where a state-action pair $(s, a)$ becomes known if its total number of visits exceeds $\alpha D_{\mathcal{G}_m} S \log(D_{\mathcal{G}_m} SA/\delta)$ for some constant $\alpha > 0$); (iii) the current episode ends, i.e., the a goal state in $\mathcal{G}_m$ is reached; (iv) the current epoch ends, i.e., the number of visits to some state-action pair is doubled. Finally, we denote by $H_m$ the length of each interval $m$, by $M$ the total number of intervals

---

[11]Note that the SSP formulation can easily handle multiple goal states. To justify this statement, we make explicit an SSP instance with single goal state that is strictly equivalent to the SSP instance $M_j$ at hand with multiple goals $\mathcal{G}_j$. To do so, we introduce an artificial terminal state $\lambda$ and define the SSP-MDP $Q_j$ with $\mathcal{S} \cup \{\lambda\}$ states (the non-goals are $\mathcal{S}$ while the unique goal is $\lambda$). Its transition dynamics $q_j$ is defined as follows: $q_j(\lambda|\lambda, a) = 1, \forall s \notin \mathcal{G}_j, q_j(s'|s, a) = p_j(s'|s, a)$, and $\forall s \in \mathcal{G}_j, q_j(\lambda|s, a) = 1$. Its cost function is set to the original costs $c_j$ for states not in $\mathcal{G}_j$, and to 0 (or equivalently any constant) for states in $\mathcal{G}_j$, and finally to 0 for the terminal state $\lambda$. This construction mirrors the one proposed by Bertsekas in the lecture `https://web.mit.edu/dimitrib/www/DP_Slides_2015.pdf` (page 25). Note that the SSP instance $M_j$ with multiple goal states $\mathcal{G}_j$ is equivalent to the single-goal SSP instance $Q_j$. The artificial terminal state $\lambda$ is not formally necessary; it justifies why having multiple goal states is well-defined from an analysis point of view.

and by $T_M := \sum_{m=1}^{M} H_m$ the total time steps. As such, $T_M$ amounts to the sample complexity that we seek to bound. Note that the goal states $\mathcal{G}_m$ are measurable at the beginning of the attempt $m$. Hence we can extend the reasoning of [45, App. B.2.7 & B.2.8] to varying goal states using the decomposition described above. Assuming throughout that the high-probability events hold, we get[12]

$$T_M = \widetilde{O}\left( \sum_{m \in \mathcal{M}^{(iii)}} D_{\mathcal{G}_m} + S\sqrt{A}\sqrt{\sum_{m=1}^{M} D_{\mathcal{G}_m}^2 + DS^2A} \right), \tag{12}$$

where $\mathcal{M}^{(iii)}$ is defined as the set of intervals that end according to condition (iii). We now proceed with the following decomposition, which is analogous to [45, Observation 4.1]

$$\sum_{m=1}^{M} D_{\mathcal{G}_m}^2 \leq \sum_{m:H_m \geq D_{\mathcal{G}_m}} D_{\mathcal{G}_m}^2 + \sum_{m:H_m < D_{\mathcal{G}_m}} D_{\mathcal{G}_m}^2.$$

Using that $D_{\mathcal{G}_m} \leq D$, the first term can be bounded as

$$\sum_{m:H_m \geq D_{\mathcal{G}_m}} D_{\mathcal{G}_m}^2 \leq D \sum_{m:H_m \geq D_{\mathcal{G}_m}} D_{\mathcal{G}_m} \leq D \sum_{m:H_m \geq D_{\mathcal{G}_m}} H_m \leq D \sum_{m=1}^{M} H_m = DT_M.$$

As for the second term, we observe that it removes intervals ending under the condition (i) and thus only accounts for intervals ending under the conditions (ii), (iii) or (iv). We now perform the following key partition of intervals: each interval is categorized depending on the first goal state that ends up being reached at the end or after the considered interval. We call this goal state the *retrospective goal state of the interval*. This retrospective categorization of intervals can be performed since it does not appear at an algorithmic level, but only appears at an analysis-level after Eq. 12 is obtained, in order to simplify it. For any interval $m$, we denote by $s_m$ its retrospective goal. Likewise, let us denote by $M_s$ (resp. $\mathcal{M}_s$) the number (resp. the set) of intervals with retrospective goal state $s$. Finally, for any $j \in \{ii, iii, iv\}$, we denote we denote by $M^{(j)}$ (resp. the $\mathcal{M}^{(j)}$) the number (resp. the set) of intervals that end according to condition $(j)$, and by $M_s^{(j)}$ (resp. $\mathcal{M}_s^{(j)}$) the number (resp. the set) of intervals with retrospective goal state $s$ that end according to condition $(j)$. We can now write

$$\sum_{m:H_m < D_{\mathcal{G}_m}} D_{\mathcal{G}_m}^2 = \sum_{m \in \mathcal{M}^{(ii)}} D_{\mathcal{G}_m}^2 + \sum_{m \in \mathcal{M}^{(iii)}} D_{\mathcal{G}_m}^2 + \sum_{m \in \mathcal{M}^{(iv)}} D_{\mathcal{G}_m}^2 = \sum_{j \in \{ii,iii,iv\}} \sum_{m \in \mathcal{M}^{(j)}} D_{\mathcal{G}_m}^2.$$

Now, for any $j \in \{ii, iii, iv\}$,

$$\begin{aligned}
\sum_{m \in \mathcal{M}^{(j)}} D_{\mathcal{G}_m}^2 = \sum_{m \in \mathcal{M}^{(j)}} \left( \sum_{s \in \mathcal{S}} \mathbb{1}_{\{s_m = s\}} \right) D_{\mathcal{G}_m}^2 &= \sum_{s \in \mathcal{S}} \sum_{m \in \mathcal{M}_s^{(j)}} D_{\mathcal{G}_m}^2 \\
&\overset{(a)}{\leq} \sum_{s \in \mathcal{S}} \sum_{m \in \mathcal{M}_s^{(j)}} D_s^2 \\
&= \sum_{s \in \mathcal{S}} M_s^{(j)} D_s^2,
\end{aligned}$$

where inequality (a) comes from Lem. 6 stated later. Moreover, we have

$$M_s^{(ii)} = \widetilde{O}(D_s S^2 A); \qquad M_s^{(iii)} \leq b(s); \qquad M^{(iv)} \leq 2SA \log(T_M).$$

While the first and third bounds above are similar to those considered in [45], the key difference lies in the second bound, which leverages that the number of intervals that end in the goal state $s$ is, by definition of our problem, upper bounded by the number of samples required at state $s$, i.e., $b(s)$. All in all, this implies that

$$\sum_{m:H_m < D_{\mathcal{G}_m}} D_{\mathcal{G}_m}^2 \leq \widetilde{O}\left( \sum_{s \in \mathcal{S}} D_s^3 S^2 A \right) + \sum_{s \in \mathcal{S}} b(s) D_s^2 + \widetilde{O}(D^2 SA).$$

---

[12]The intuition behind Eq. 12 comes from the Cauchy-Schwarz inequality. For instance, let us consider the objective of bounding the quantity $Y := \sum_m x_m \sqrt{y_m}$, where the $(x_m)$ correspond to the SSP-diameters considered at each interval $m$ and the $(y_m)$ are the summands whose sums are bounded by [45, Lem. B.16]. In the latter work, denoting by $\overline{x}$ the common upper bound on the $(x_m)$, the analysis yields $Y \leq \overline{x} \sum_m \sqrt{y_m} \leq \overline{x}\sqrt{M}\sqrt{\sum_m y_m}$. In contrast, our setting requires to perform the tighter inequality $Y \leq \sqrt{\sum_m x_m^2}\sqrt{\sum_m y_m}$.

Moreover, in a similar manner as above, we bound the first term of Eq. 12 as follows

$$\sum_{m \in \mathcal{M}^{(iii)}} D_{\mathcal{G}_m} = \sum_{s \in \mathcal{S}} M_s^{(iii)} D_s \leq \sum_{s \in \mathcal{S}} D_s b(s).$$

Putting everything together back into Eq. 12 and simplifying using the subadditivity of the square root, we get

$$T_M = \widetilde{O}\left( \sum_{s \in \mathcal{S}} D_s b(s) + D S^2 A + S\sqrt{A D T_M} + S\sqrt{A}\sqrt{\sum_{s \in \mathcal{S}} b(s) D_s^2} + S^2 A \sum_{s \in \mathcal{S}} D_s^{3/2} \right).$$

Using that $x \leq c_1 \sqrt{x} + c_2$ implies $x \leq (c_1 + \sqrt{c_2})^2$ for $c_1 \geq 0$ and $c_2 \geq 0$, we obtain

$$T_M = \widetilde{O}\left( \left[ S\sqrt{DA} + \sqrt{\sum_{s \in \mathcal{S}} D_s b(s)} + \sqrt{S\sqrt{A}\sqrt{\sum_{s \in \mathcal{S}} b(s) D_s^2} + \sqrt{S^2 A \sum_{s \in \mathcal{S}} D_s^{3/2}}} \right]^2 \right). \quad (13)$$

We now apply the Cauchy-Schwarz inequality to simplify the third summand

$$S\sqrt{A}\sqrt{\sum_{s \in \mathcal{S}} b(s) D_s^2} \leq \sum_{s \in \mathcal{S}} \sqrt{S^2 A D_s}\sqrt{D_s b(s)} \leq \sqrt{\sum_{s \in \mathcal{S}} D_s b(s)}\sqrt{S^2 A \sum_{s \in \mathcal{S}} D_s}.$$

Let us introduce $x := \sqrt{\sum_{s \in \mathcal{S}} D_s b(s)}$ and $y := \sqrt{S^2 A \sum_{s \in \mathcal{S}} D_s^{3/2}}$. Plugging the simplifications into Eq. 13 finally yields with probability at least $1 - \delta$ that $T_M = \widetilde{O}\left( \left(x + \sqrt{xy} + y\right)^2 \right) = \widetilde{O}\left( (x + y)^2 \right) = \widetilde{O}(x^2 + y^2)$. Since $T_M$ amounts to the sample complexity, we get the desired bound of Eq. 7, which reads

$$\mathcal{C}(\text{GOSPRL}, b, \delta) = \widetilde{O}\left( \sum_{s \in \mathcal{S}} D_s b(s) + S^2 A \sum_{s \in \mathcal{S}} D_s^{3/2} \right).$$

**Lemma 6.** *For any set of goals $\mathcal{G} \subsetneq \mathcal{S}$, we introduce the meta SSP-diameter $D_{\mathcal{G}} := \max_{s \in \mathcal{S} \setminus \mathcal{G}} \min_{\pi} \mathbb{E}[\tau_\pi(s \to \mathcal{G})]$, where we define $\tau_\pi(s \to \mathcal{G}) := \min\{t \geq 0 : s_{t+1} \in \mathcal{G} \mid s_1 = s, \pi\}$. Then we have*

$$D_{\mathcal{G}} \leq \min_{s \in \mathcal{G}} D_s.$$

*Proof.* For any $g \in \mathcal{G}$, $s \in \mathcal{S} \setminus \mathcal{G}$ and policy $\pi$, we have $\mathbb{E}[\tau_\pi(s \to \mathcal{G})] \leq \mathbb{E}[\tau_\pi(s \to g)]$. In particular, this implies that for any $g \in \mathcal{G}$, $D_{\mathcal{G}} \leq D_g$, which immediately gives the result. $\qquad\square$

### C.2  From Corollary 1 to Theorem 1

We now consider the general case of possibly action-dependent and time-dependent sampling requirements.

**State-action requirements.** First, GOSPRL can be easily extended from state requirements $b(s)$ to state-action requirements $b(s, a)$. Indeed, the only difference between these two settings occurs w.r.t. which action the algorithm takes at the end of a given episode (i.e., when a sought-after goal state is reached): for state-action requirements, any under-sampled action is taken (see footnote 10 for details). Bound-wise, the number of times where this scenario occurs is at most $B$ (since there are at most $B$ episodes), hence the guarantee from Cor. 1 is unaffected whatever the action executed once a goal state is reached.

**Adaptive requirements.** GOSPRL can be also easily extended to requirements $(b_t(s, a))_{t \geq 1}$ that vary over time, where $b_t$ may be chosen adaptively depending on the samples observed so far (i.e., $b_t$ is measurable w.r.t. the filtration up to time $t$). Indeed, the important property required in the derivations of App. C.1 that both the starting state and the goal states should be measurable (i.e., known and fixed) at the beginning of each episode still holds. As such, the sample complexity result of Cor. 1 can be naturally extended by defining $B_\tau := \sum_{s,a} b_\tau(s, a)$, where $\tau$ is the first (random) time step when all the sampling requirements are met. In order for the sample complexity to remain bounded, a sufficient condition is Asm. 2. In particular, considering the sequence $b_t(s, a)$ to be upper bounded by a fixed threshold $\bar{b}(s, a)$ for each $(s, a)$, the bound from Cor. 1 trivially holds with $\overline{B} := \sum_{s,a} \bar{b}(s, a)$.

## C.3 Remark

Notice that the "comparator" we are using in the definition of the regret in Eq. 8 may not be the "global" optimum in terms of sample complexity. Indeed, the optimal sequence of strategies would result in a non-stationary policy $\pi_{\mathcal{C}}^{\star} \in \arg\min_{\pi} \mathcal{C}(\pi, b, \delta)$. Yet in our analysis, we compare the algorithmic performance with the larger quantity $\sum_{j=1}^{J} \min_{\pi} V_{\mathcal{G}_j}^{\pi}(\underline{s}_j)$, which corresponds to "greedily" minimizing each time to reach an under-sampled state in a sequential fashion. This highlights that GOSPRL does not *track* any optimal sampling allocation or distribution (i.e., it does not seek to "imitate" $\pi_{\mathcal{C}}^{\star}$), insofar as it discards the effect of traversing other states while reaching an undersampled goal state. While this means that some areas of the state space may be oversampled, GOSPRL is able to devote its full attention to the objective of minimizing the total sample complexity, instead of being mindful to avoid certain areas of the state space which it has already visited. We argue that this is what results in the appealing sample complexity of GOSPRL, whereas other techniques specifically designed to track distributions (via e.g., the Frank-Wolfe algorithmic scheme) struggle to minimize the sample complexity, as explained in Sect. 4.1 and App. G.

## D   Lower Bound

In this section, we provide three complementary results that lower bound the sample complexity of the problem of Def. 1.

① **First**, as stated in Lem. 1, we construct a simple MDP such that for any arbitrary sampling requirements $b(s)$, the (possibly non-stationary) policy minimizing the time to collect all samples has sample complexity of order $\Omega\big(\sum_{s \in \mathcal{S}} D_s b(s)\big)$. We begin with a useful result.

**Lemma 7.** *Let $q \in (0,1)$ and consider the Markov chain $M_q$ with two states $x$, $y$ whose dynamics $p_q$ are as follows: $p_q(y|x) = q$, $p_q(x|x) = 1 - q$ and $p_q(x|y) = 1$. Then $M_q$ is communicating with diameter $D_q := \frac{1}{q}$. Moreover, denote by $T_B$ the (random) time of the $B$-th visit to state $y$ starting from any state, and assume that $B \geq 5$. Then with probability at least $\frac{1}{2}$, we have $T_B \geq \frac{B}{2q} + B = \frac{BD_q}{2} + B$.*

*Proof.* Introduce $X := \sum_{i=1}^{n} X_i$ where $X_i \sim Ber(q)$ (i.e., it follows a Bernoulli with parameter $q$) and we set $n := \frac{B}{2q}$. We have $\mathbb{E}[X] = nq = \frac{B}{2}$. Moreover, the Chernoff inequality entails that

$$\mathbb{P}(X \geq B) = \mathbb{P}(X \geq 2\mathbb{E}[X]) \leq \exp\left(-\frac{\mathbb{E}[X]}{3}\right) = \exp\left(-\frac{B}{6}\right) \leq \frac{1}{2},$$

where the last inequality holds whenever $B \geq 6\log(2)$. Note that the random variable $T_B$ follows a negative binomial distribution for which each success accounts for two time steps instead of one. This means that with probability at least $\frac{1}{2}$,

$$T_B \geq n + B = \frac{B}{2q} + B = \frac{BD_q}{2} + B.$$

$\square$

Let us now consider a state space $\mathcal{S} := \{s_1, \ldots, s_S\}$ and arbitrary sampling requirements $b : \mathcal{S} \to \mathbb{N}$. We construct a wheel MDP with state space $\mathcal{S} \cup \{s_0\}$, where $s_0$ is the starting center state. There are $A = S$ actions available and the dynamics $p$ are defined w.r.t. a set $(\varepsilon_i) \in (0,1)^S$ such that $\forall i \in [S]$, $p(s_i|s_0, a_i) = \varepsilon_i$, $p(s_0|s_0, a_i) = 1 - \varepsilon_i$, and for every action $a$, $p(s_0|s_i, a) = 1$. Note that by having such $A = S$ actions, the attempts to collect relevant samples are independent, in the sense that at any $s \in \mathcal{S}$, the learner cannot rely on the attempts performed for the other states $s' \neq s$. Let us assume that $b(s) \geq 6\log(2S)$. From Lem. 7, for any state $s \in \mathcal{S}$, with probability $1 - \frac{1}{2S}$, the time needed to collect $b(s)$ samples from state $s$ is lower bounded by $\frac{b(s)}{2\varepsilon_i} + b(s)$, and furthermore we have $D_s = \frac{1}{\varepsilon_i} + 1$. Taking a union bound over the $S$ states in $\mathcal{S}$ means that with probability at least $\frac{1}{2}$, the time to collect the required samples is lower bounded by $\sum_{s \in \mathcal{S}} \frac{b(s)(D_s - 1)}{2} + b(s)$.

② **Second**, we show that the family of worst-case MDPs is relatively large. In fact, for any MDP with diameter $D$, we can perform a minor change to its dynamics without affecting the overall diameter

and show that when the sampling requirements are concentrated in a single state, any policy would take at least $\Omega(BD)$ steps to collect all the $B$ samples. More specifically, there exists a class $\mathbb{C}$ of MDPs such that, for each MDP in $\mathbb{C}$, there exists a requirement function $b$ and a finite threshold (that depends on the considered MDP) such that the $\Omega(BD)$ lower bound holds whenever $B$ exceeds this threshold. The class $\mathbb{C}$ effectively encompasses a large number of environments: indeed, take *any* MDP $M$, then we can find an MDP $M'$ in $\mathbb{C}$ such that $M$ and $M'$ differ in their transitions *only* at one state and have the same diameter. Formally, we have the following statement (proof in App. D.1).

**Lemma 8.** *Fix any positive natural numbers $S$, $A$ and $D$, and any MDP $M$ with $S = |\mathcal{S}|$ states, $A = |\mathcal{A}|$ actions and diameter $D$. There exists a modification of the transitions of $M$ at only one state which yields an MDP $M'$ with the same diameter $D$, and there exists a finite integer $W_{\mathfrak{A},M',\delta}$ (depending on $\mathfrak{A}$, $M'$) such that for any total requirement $B \geq W_{\mathfrak{A},M',\delta}$, there exists a function $b^\dagger : \mathcal{S} \to \mathbb{N}$ with $\sum_{s \in \mathcal{S}} b(s) = B$, such that, for any arbitrary starting state, the optimal non-stationary policy $\mathfrak{A}^\star$ needs $\mathcal{C}(\mathfrak{A}^\star, b^\dagger)$ time steps to collect the desired samples in the modified MDP $M'$, where*

$$\mathbb{P}\left(\mathcal{C}(\mathfrak{A}^\star, b^\dagger) > \frac{(B-1)D}{2}\right) \geq \frac{1}{2}.$$

③ **Third**, we note that both results above do not take into account the added difficulty for the agent to have to deal with a learning process. To do so, we can draw inspiration from the lower bound on the expected regret for learning in an SSP problem derived in [45]. Indeed, let us consider a environment $M$ with one state $\overline{s}$ in which all the required samples are concentrated, i.e., $b := B\mathbb{1}_{\overline{s}}$ with $B \geq SA$. The $S-1$ other states $s$ each contain a special action $a_s^\star$. The transition dynamics $p$ are defined as follows: $p(\overline{s}|s, a_s^\star) = \frac{1}{D_{\overline{s}}}, p(s|s, a_s^\star) = 1 - \frac{1}{D_{\overline{s}}}, p(\overline{s}|s, a) = \frac{1-\nu}{D_{\overline{s}}}, p(s|s, a) = 1 - \frac{1-\nu}{D_{\overline{s}}}$ for any other action $a \in \mathcal{A} \setminus \{a_s^\star\}$, and finally $p(s|\overline{s}, a) = \frac{1}{S-1}$ for any action $a \in \mathcal{A}$, with $\nu := \sqrt{(S-1)AB}/64$. Recall that $D_{\overline{s}}$ is the SSP-diameter of state $\overline{s}$. The communicating, non-episodic structure of $M$ naturally mimics the interaction of an agent with an SSP problem with goal state $\overline{s}$. Denoting by $\mathcal{C}(\mathfrak{A}, b)$ the (random) time required by any algorithm $\mathfrak{A}$ to collect the $b$ sought-after samples, we obtain from [45, Thm. 2.7] that

$$\mathbb{E}[\mathcal{C}(\mathfrak{A}, b = B\mathbb{1}_{\overline{s}})] \geq \phi(B) := \underbrace{(D_{\overline{s}} + 1)B}_{:=\phi_1(B)} + \underbrace{\frac{1}{1024}D_{\overline{s}}\sqrt{(S-1)AB}}_{:=\phi_2(B)}$$

$$= \sum_{s \in \mathcal{S}}\left((D_s + 1)b(s) + \frac{1}{1024}D_s\sqrt{(S-1)Ab(s)}\right).$$

This lower bound on the expected time to collect the samples implies in particular that no algorithm can meet the sampling requirements in less than $\widetilde{O}(\phi(B))$ time steps with high probability. Importantly, note that this result is not contradictory with Thm. 1. Indeed, as fleshed out in the proof in App. C, the upper bound of Thm. 1 actually contains such a square root term $\phi_2(B)$, yet it is subsumed in the final bound by either the main-order term in $\sum_s b(s)D_s$ or the lower-order term constant w.r.t. $B$ (see Eq. 11). We can decompose $\phi(B)$ in two factors: the second term $\phi_2(B)$ comes from the learning process of trying to match the behavior of the optimal policy, while the first term $\phi_1(B)$ stems from the need to navigate through the environment as opposed to the generative model assumption (as such, it is incurred even if the optimal policy is deployed from the start). Part ② of this section actually shows that such a term $\phi_1(B)$ is unavoidable in multiple MDPs.

### D.1 Proof of Lemma 8

Here we give the proof of Lem. 8. For any positive natural numbers $S$, $A$, $D$, we consider any MDP $M$ with $S$ states, $A$ actions and diameter $D$. We consider

$$(\underline{s}, \overline{s}) \in \arg\max_{s \neq s' \in \mathcal{S}}\left\{\min_{\pi \in \Pi^{\text{SD}}} \mathbb{E}[\tau_\pi(s \to s')]\right\}.$$

We modify the transition structure of $M$, so that $p(\underline{s}|\overline{s}, a) = 1$ for all actions $a \in \mathcal{A}$. Note that the diameter is not affected by this operation. Throughout, whatever the value of $B$, we will consider the

following sampling requirements: $b(s) := B\mathbb{1}_{\{s=\overline{s}\}}$. We denote by $s_0 \in \mathcal{S}$ the arbitrary starting state of the learning process.

Consider any learning algorithm $\mathfrak{A}$. We denote by $\pi$ the (possibly non-stationary) policy that is executed by $\mathfrak{A}$. In virtue of Asm. 1, we can naturally (and without loss of generality) restrict our attention to a policy $\pi$ whose expected hitting time to $\overline{s}$ is finite starting from any state in $\mathcal{S}$ — we denote by $\overline{\mu}_\pi$ such an upper bound. We denote by $T_\pi^{(i)}$ the random time required by policy $\pi$ to collect the $i$-th sample at state $\overline{s}$, starting from $s_0$ if $i = 1$ or from $\underline{s}$ if $2 \leq i \leq B$.

**Lemma 9.** *The $(T_\pi^{(i)})_{2 \leq i \leq B}$ are i.i.d. sub-exponential random variables whose expectation satisfies $\mu_\pi := \mathbb{E}\left[T_\pi^{(i)}\right] \geq D$ for all $2 \leq i \leq B$.*

*Proof.* Consider the SSP problem with unitary costs, starting state $\underline{s}$ and zero-cost, absorbing terminal state $\overline{s}$. According to [10], Asm. 1 and the fact that the costs are all positive guarantee that the optimal value function of this SSP problem is achieved by a stationary deterministic policy. This implies that $\min_{\pi' \in \Pi^{\text{SD}}} \mathbb{E}[\tau_{\pi'}(\underline{s} \to \overline{s})] \leq \mathbb{E}[\tau_\pi(\underline{s} \to \overline{s})]$, and thus by definition of $D$ and $\mu_\pi$, we get the inequality $D \leq \mu_\pi$. There remains to prove the sub-exponential nature of the random variable $T_\pi$. For any $\lambda \in \mathbb{R}$, we have

$$\mathbb{E}\left[e^{\lambda(T_\pi - \mu_\pi)}\right] = e^{-\lambda\mu_\pi}\mathbb{E}\left[\sum_{n=0}^{+\infty}\frac{1}{n!}\lambda^n T_\pi^n\right] = e^{-\lambda\mu_\pi}\sum_{n=0}^{+\infty}\frac{1}{n!}\lambda^n\mathbb{E}[T_\pi^n] \leq 2e^{-\lambda\mu_\pi}\sum_{n=0}^{+\infty}\frac{1}{n!}n^n(\lambda\overline{\mu}_\pi)^n,$$

where the last inequality comes from [50, Lem. 15], which can be applied to bound the moments $\mathbb{E}[T_\pi^n] \leq 2(n\overline{\mu}_\pi)^n$, since the random variable $T_\pi$ satisfies $\mathbb{E}[T_\pi(s \to \overline{s})] \leq \overline{\mu}_\pi$ for all $s \in \mathcal{S}$ by definition of $\overline{\mu}_\pi$. From Lem. 10, the series above converges whenever $|\lambda| < \frac{1}{e\overline{\mu}_\pi}$. This proves that $T_\pi$ is sub-exponential according to the second condition of Def. 4. $\qquad\square$

**Lemma 10.** *The series $\displaystyle\sum_{n=0}^{+\infty}\frac{n^n}{n!}x^n$ converges absolutely for all $|x| < \frac{1}{e}$.*

*Proof.* Introduce the summand of the series $a_n(x) := \frac{n^n}{n!}x^n$. We then have

$$\frac{a_{n+1}(x)}{a_n(x)} = \frac{n!}{(n+1)!}\frac{(n+1)^{n+1}}{n^n}x = \left(1 + \frac{1}{n}\right)^n x \xrightarrow[n\to+\infty]{} ex.$$

Hence, for any $|x| < \frac{1}{e}$, we have $|\frac{a_{n+1}(x)}{a_n(x)}| < 1$, which means from d'Alembert's ratio test that the series converges absolutely. $\qquad\square$

Since $T_\pi$ is sub-exponential, from Def. 4, there exists a pair $(\sigma_\pi, \theta_\pi)$ of finite positive parameters that verifies

$$\mathbb{E}\left[e^{\lambda(T_\pi - \mu_\pi)}\right] \leq e^{\frac{\sigma_\pi^2\lambda^2}{2}} \quad \text{for all } |\lambda| < \frac{1}{\theta_\pi}.$$

We now apply the concentration inequality for sub-exponential random variables stated in Lem. 11.

$$\forall y > \frac{\sigma_\pi^2}{\theta_\pi}, \quad \mathbb{P}\left(\sum_{i=2}^{B}T_\pi^{(i)} \leq \mu_\pi(B-1) - y\right) \leq \exp\left(-\frac{y}{2\theta_\pi}\right).$$

We now fix the integer

$$W_\pi := 1 + 2\max\left\{\left\lceil\frac{\theta_\pi}{\mu_\pi}\right\rceil, \left\lceil\frac{\sigma_\pi^2}{\theta_\pi\mu_\pi}\right\rceil\right\}.$$

Consider any total sampling requirement $B \geq W_\pi$. Then setting $y := \frac{\mu_\pi(B-1)}{2} > \frac{\sigma_\pi^2}{\theta_\pi}$ yields

$$\mathbb{P}\left(\sum_{i=2}^{B}T_\pi^{(i)} \leq \frac{\mu_\pi(B-1)}{2}\right) \leq \exp\left(-\frac{\mu_\pi(B-1)}{4\theta_\pi}\right) \leq \frac{1}{2},$$

since we have $B \geq \frac{4\theta_\pi}{\mu_\pi} \log(2) + 1$. This implies that with probability at least $\frac{1}{2}$,

$$\sum_{i=1}^{B} T_\pi^{(i)} \geq \sum_{i=2}^{B} T_\pi^{(i)} > \frac{\mu_\pi(B-1)}{2} \geq \frac{(B-1)D}{2},$$

where the last inequality stems from Lem. 9. As a result, there exists a finite integer $W_{\pi,\delta}$ (depending on $\pi$ and the environment at hand) such that, for any total sampling requirement $B \geq W_\pi$, the algorithm $\mathfrak{A}$ that executes policy $\pi$ verifies

$$\mathbb{P}\left(\mathcal{C}(\mathfrak{A}, B\mathbb{1}_{\{\bar{s}\}}) > \frac{(B-1)D}{2}\right) \geq \frac{1}{2},$$

which gives the proof of Lem. 8.

We recall here the definition of sub-exponential random variables.

**Definition 4** (57)**.** *A random variable $X$ with mean $\mu < +\infty$ is said to be sub-exponential if one of the following equivalent conditions is satisfied:*

1. *(Laplace transform condition) There exists $(\sigma, \theta) \in \mathbb{R}^+ \times \mathbb{R}^{+\star}$ such that, for all $|\lambda| < \frac{1}{\theta}$,*

$$\mathbb{E}\left[e^{\lambda(X-\mu)}\right] \leq e^{\frac{\sigma^2\lambda^2}{2}}.$$

2. *There exists $c_0 > 0$ such that $\mathbb{E}\left[e^{\lambda(X-\mu)}\right] < +\infty$ for all $|\lambda| \leq c_0$.*

*For any pair $(\sigma, \theta)$ satisfying condition 1, we write $X \sim \text{SUBEXP}(\sigma, \theta)$.*

We finally recall a concentration inequality satisfied by sub-exponential random variables.

**Lemma 11** (57)**.** *Let $(X_i)_{1 \leq i \leq n}$ be a collection of independent sub-exponential random variables such that for all $i \in [n]$, $X_i \sim \text{SUBEXP}(\sigma_i, \theta_i)$ and $\mu_i := \mathbb{E}[X_i]$. Set $\sigma := \sqrt{\frac{\sum_{i=1}^n \sigma_i^2}{n}}$ and $\theta := \max_{i \in [n]}\{\theta_i\}$. The following concentration inequalities hold for any $t \geq 0$,*

$$\mathbb{P}\left(\sum_{i=1}^{n} X_i - \sum_{i=1}^{n} \mu_i \geq t\right) \leq \begin{cases} e^{-\frac{t^2}{2n\sigma^2}} & \text{if } 0 \leq t \leq \frac{\sigma^2}{\theta} \\ e^{-\frac{t}{2\theta}} & \text{if } t > \frac{\sigma^2}{\theta} \end{cases},$$

$$\mathbb{P}\left(\sum_{i=1}^{n} X_i - \sum_{i=1}^{n} \mu_i \leq -t\right) \leq \begin{cases} e^{-\frac{t^2}{2n\sigma^2}} & \text{if } 0 \leq t \leq \frac{\sigma^2}{\theta} \\ e^{-\frac{t}{2\theta}} & \text{if } t > \frac{\sigma^2}{\theta} \end{cases}.$$

# E    GOSPRL Beyond the Communicating Setting

Sect. 3.1 and App. D demonstrate that the diameter $D$ and/or the SSP-diameters dictate the performance of a sampling procedure in a communicating environment. Indeed, both the GOSPRL upper bound and the worst-case lower bound contain $D$ and/or $D_s$ as a multiplicative factor w.r.t. the total sampling requirement $B$. However, in many environments, there may exist some states that are hard to reach, or plainly impossible to reach. In that case, the diameter is prohibitively large and even possibly infinite, thus rendering the sample complexity guarantee of Thm. 1 vacuous. To circumvent this issue, a desirable property of the algorithm would be the ability to assess online the "feasibility" of the sampling requirements, by discarding states that are indeed too difficult to reach. For ease of exposition, we consider throughout App. E the special case of time- and action-independent sampling requirements $b : \mathcal{S} \to \mathbb{N}$ (as explained in App. C.2 the extension to the general case of adaptive action-dependent sampling requirements follows straightforwardly).

Formally, we consider any environment that need not be communicating (i.e., it may not satisfy Asm. 1). The learning agent receives as input an integer parameter $L \geq 1$, which acts as a reachability threshold that partitions the state space between the states from which we expect sample collection and those that we categorize as too difficult to reach. Specifically, given a sampling requirement $b : \mathcal{S} \to \mathbb{N}$, the desiderata of the agent is to minimize the time it requires, for each state $s \in \mathcal{S}$, to i) either collect the $b(s)$ samples, ii) or discard the sample collection at state $s$ only if there exists a state (accessible from the starting state) that cannot reach $s$ within $L$ steps in expectation. In other words, we do not allow for samples to be discarded if the state is actually below the reachability threshold $L$. We introduce the following new definition of the sample complexity.

**Definition 5.** *Given a reachability threshold $L \geq 1$, sampling requirements $b : \mathcal{S} \to \mathbb{N}$, starting state $s_0 \in \mathcal{S}$ and a confidence level $\delta \in (0, 1)$, the sample complexity of a learning algorithm $\mathfrak{A}$ is defined as*

$$\mathcal{C}(\mathfrak{A}, b, \delta, L, s_0) := \min \left\{ t > 0 : \mathbb{P}\Big(\forall s \in \mathcal{S}_L, \ N_t(s) \geq b(s) \wedge I_{\mathfrak{A}}(t) = 1\Big) \geq 1 - \delta \right\},$$

*where $\mathcal{S}_L := \{s \in \mathcal{S} : \max_{\{y \in \mathcal{S}: D_{s_0 y} < +\infty\}} D_{ys} \leq L\}$ and where $I_{\mathfrak{A}}(t)$ corresponds to a Boolean equal to $1$ if the algorithm $\mathfrak{A}$ considers at time $t$ that none of the states that remain to be sampled (if there remains any) belong to $\mathcal{S}_L$.[13]*

**Algorithm GOSPRL-L.** We now propose a simple adaptation of GOSPRL to handle this setting, and call the corresponding algorithm GOSPRL-L since it receives as input a reachability threshold $L$. We split time in *episodes* indexed by $j$, where the first episode begins at the first time step and the $j$-th episode ends when the $j$-th desired sample is collected. From Thm. 1 we know that in a communicating environment with diameter $D$, there exists an absolute constant $\alpha > 0$ (here we exclude logarithmic terms for ease of exposition) such that with probability at least $1 - \delta$, after any $j$ episodes (i.e., after the $j$-th desired sample is collected), $T_j$ the (total) time step at the end of the $j$ episodes is upper bounded as follows

$$T_j \leq \alpha j D + \alpha j D^{3/2} S^2 A.$$

The key idea is to run GOSPRL and stop its execution if its total duration at some point exceeds a certain threshold depending on $L$ and the current episode. Specifically, in the $j$-th episode, this threshold is set to $\Phi(j) := \alpha j L + \alpha j L^{3/2} S^2 A$. If the accumulated duration never exceeds the threshold, the algorithm is naturally run until all the sampling requirements are met.

**Lemma 12.** *Consider any reachability threshold $L \geq 1$, starting state $s_0 \in \mathcal{S}$, confidence level $\delta \in (0, 1)$ and sampling requirements $b : \mathcal{S} \to \mathbb{N}$, with $B = \sum_{s \in \mathcal{S}} b(s)$. Then running the algorithm GOSPRL-L in any environment yields a sample complexity that can be upper bounded as*

$$\mathcal{C}(\text{GOSPRL-L}, b, \delta, L, s_0) = \widetilde{O}\Big(BL + L^{3/2} S^2 A\Big).$$

*Proof.* The result is obtained by performing a *reductio ad absurdum* reasoning. We initially make the assumption $\mathcal{H}$ that for all episodes $j \geq 1$, we have $D_{\mathcal{G}_j} \leq L$, where we recall that $D_{\mathcal{G}_j}$ is the SSP-diameter of the goal states $\mathcal{G}_j$ considered during episode $j$. The condition that is checked at any time step is whether it is smaller or larger than the threshold $\Phi(j) := \alpha j L + \alpha j L^{3/2} S^2 A$, where $j$ is the current episode. ***i)*** In the first case, the total duration is always smaller (or equal) than its threshold and the algorithm performs $J$ episodes until the sampling requirements are met. Since $J \leq B$ and $\Phi$ is an increasing function, the sample complexity is bounded by $\Phi(J) \leq \Phi(B) = \widetilde{O}\big(BL + L^{3/2} S^2 A\big)$. ***ii)*** In the second case, there exists an episode $j' \geq 1$ and a time step (during that episode) which is larger than the threshold $\Phi(j')$. This implies that with probability at least $1 - \delta$, assumption $\mathcal{H}$ is wrong. Thus there exists an episode $1 \leq j \leq j'$ such that $D_{\mathcal{G}_j} > L$. Since $\mathcal{G}_{j'} \subset \mathcal{G}_j$, we have $D_{\mathcal{G}_j} \leq D_{\mathcal{G}_{j'}}$, thus $D_{\mathcal{G}_{j'}} > L$, which implies from Lem. 6 that for all $s \in \mathcal{G}_{j'}$, $D_s > L$. Hence the algorithm can terminate and confidently guarantee that none of the states that remain to be sampled belong to $\mathcal{S}_L$. Given that $j' \leq B$, the sample complexity (in the sense of Def. 5) is bounded by $\Phi(j') \leq \Phi(B) = \widetilde{O}\big(BL + L^{3/2} S^2 A\big)$. $\qquad\square$

The algorithm GOSPRL-L requires no computational overhead w.r.t. GOSPRL, as it simply tracks the total duration of GOSPRL and terminates if it exceeds a threshold depending on $L$. Under the new appropriate definition of sample complexity of Def. 5, the dependency in Thm. 1 on the possibly very large or infinite diameter $D$ is effectively replaced by the reachability threshold $L$. A large value of $L$ signifies that the sample collection is required at quite difficult-to-reach states, while a small value of $L$ keeps in check the duration of the sampling procedure.

---

[13] Why isn't $\mathcal{S}_L$ defined as $\mathcal{S}_L := \{s \in \mathcal{S} : D_s \leq L\}$? Under such a definition, in the case of a weakly communicating environment, the optimal strategy $\mathfrak{A}$ would be to set $I_{\mathfrak{A}}(t = 1) = 1$, which would yield a sample complexity of $1$, since there would exist at least one "isolated" state and hence $\mathcal{S}_L = \emptyset$. Of course, this is not the behavior we would want, as we expect the optimal strategy to perform the sample collection at states in the communicating class (starting from $s_0$), and discard the sample collection at states that are not accessible from $s_0$. This is explained in more detail in the last paragraph of this section.

Narrowing the sample collection to states in $\mathcal{S}_L$ may seem at first glance restrictive. Indeed, the presence of states in which the agent may get stuck could disrupt the learning process. However, assume for instance that we consider the canonical assumption made in episodic RL of a *resetting* environment, i.e., an environment that contains a reset action that brings the agent with probability 1 to a reference starting state $s_0$ (where here we consider that the reset action can be executed at any time step for simplicity). Then we have that $\{s \in \mathcal{S} : \min_\pi \mathbb{E}[\tau_\pi(s_0 \to s)] \le L - 1\} \subseteq \mathcal{S}_L$, which shows that numerous states can effectively belong to the set $\mathcal{S}_L$.

Finally, let us delve into the particular case of a weakly communicating MDP, whose state space $\mathcal{S}$ can be partitioned into two subspaces [43, Sect. 8.3.1]: a communicating set of states (denoted $\mathcal{S}^{\mathrm{C}}$) with each state in $\mathcal{S}^{\mathrm{C}}$ accessible — with non-zero probability — from any other state in $\mathcal{S}^{\mathrm{C}}$ under some stationary deterministic policy, and a (possibly empty) set of states that are transient under all policies (denoted $\mathcal{S}^{\mathrm{T}}$). The sets $\mathcal{S}^{\mathrm{C}}$ and $\mathcal{S}^{\mathrm{T}}$ form a partition of $\mathcal{S}$, i.e., $\mathcal{S}^{\mathrm{C}} \cap \mathcal{S}^{\mathrm{T}} = \emptyset$ and $\mathcal{S}^{\mathrm{C}} \cup \mathcal{S}^{\mathrm{T}} = \mathcal{S}$. Finally, we denote by $D^{\mathrm{C}} < +\infty$ the diameter of the communicating part of $M$ (i.e., restricted to the set $\mathcal{S}^{\mathrm{C}}$), i.e., $D^{\mathrm{C}} := \max_{s \ne s' \in \mathcal{S}^{\mathrm{C}}} \min_{\pi \in \Pi^{\mathrm{SD}}} \mathbb{E}[\tau_\pi(s \to s')] < +\infty$. Assume that the starting state $s_0$ belongs to $\mathcal{S}^{\mathrm{C}}$. We expect the optimal strategy to perform the sample collection at states in $\mathcal{S}^{\mathrm{C}}$ and discard the sample collection at states in $\mathcal{S}^{\mathrm{T}}$. This is what GOSPRL-L does if we have $\mathcal{S}_L = \mathcal{S}^{\mathrm{C}}$, i.e., whenever $D^{\mathrm{C}} \le L$. Hence, in that setting, the optimal (yet critically unknown) value of the threshold $L$ would be $D^{\mathrm{C}}$.

## F    Application: Model Estimation (MODEST)

In this section we demonstrate that GOSPRL can be readily applied to tackle the MODEST problem, as well as a "robust" variant called RMODEST, both of which are defined as follows. The agent $\mathfrak{A}$ interacts with the environment and, after $t$ time steps, it must return an estimate $\widehat{p}_{\mathfrak{A},t}$ of the transition dynamics, which naturally corresponds to the empirical average of the transition probabilities. The accuracy of the estimate and the corresponding sample complexity are evaluated as follows.

**Definition 6.** *Given an accuracy level $\eta > 0$ and a confidence level $\delta \in (0,1)$, the MODEST and RMODEST sample complexity of an online learning algorithm $\mathfrak{A}$ are defined as*

$$\mathcal{C}_{\mathrm{MODEST}}(\mathfrak{A}, \eta, \delta) := \min \left\{ t > 0 : \mathbb{P}\big(\forall (s,a) \in \mathcal{S} \times \mathcal{A}, \; \|\widehat{p}_{\mathfrak{A},t}(\cdot|s,a) - p(\cdot|s,a)\|_1 \le \eta \big) \ge 1 - \delta \right\},$$

$$\mathcal{C}_{\mathrm{RMODEST}}(\mathfrak{A}, \eta, \delta) := \min \left\{ t > 0 : \mathbb{P}\big(\forall (s',s,a) \in \mathcal{S}^2 \times \mathcal{A}, \; |\widehat{p}_{\mathfrak{A},t}(s'|s,a) - p(s'|s,a)| \le \eta \big) \ge 1 - \delta \right\},$$

*where $\widehat{p}_{\mathfrak{A},t}$ is the estimate (i.e., empirical average) of the transition dynamics $p$ after $t$ time steps.*

We have the following sample complexity guarantees.

**Lemma 13.** *Instantiating GOSPRL with two different sequences of sampling requirements yields respectively*

$$\mathcal{C}_{\mathrm{RMODEST}}(\mathrm{GOSPRL}, \eta, \delta) = \widetilde{O}\Big( \frac{DSA}{\eta^2} + D^{3/2}S^2A \Big),$$

$$\mathcal{C}_{\mathrm{MODEST}}(\mathrm{GOSPRL}, \eta, \delta) = \widetilde{O}\Big( \frac{D\Gamma SA}{\eta^2} + \frac{DS^2A}{\eta} + D^{3/2}S^2A \Big).$$

*Proof.* We first focus on the RMODEST objective with desired accuracy level $\eta$. From Def. 6, we would like that, for any state-action pair $(s,a)$ and next state $s'$, the following condition holds:

$$|\widehat{p}_t(s'|s,a) - p(s'|s,a)| \le \eta. \tag{14}$$

From the empirical Bernstein inequality (see e.g., 4, 22), we have with probability at least $1 - \delta$, for any time step $t \ge 1$ and for any state-action pair $(s,a)$ and next state $s'$,

$$|\widehat{p}_t(s'|s,a) - p(s'|s,a)| \le 2\sqrt{\frac{\widehat{\sigma}_t^2(s'|s,a)}{N_t^+(s,a)} \log\Big( \frac{2SAN_t^+(s,a)}{\delta} \Big)} + \frac{6 \log\Big( \frac{2SAN_t^+(s,a)}{\delta} \Big)}{N_t^+(s,a)}, \tag{15}$$

where $N_t^+(s,a) := \max\{1, N_t(s,a)\}$ and where the $\widehat{\sigma}_t^2$ are the population variance of transitions, i.e., $\widehat{\sigma}_t^2(s'|s,a) := \widehat{p}_t(s'|s,a)(1 - \widehat{p}_t(s'|s,a))$. Let us now define, for any $X, Y \ge 0$, the quantity

$$\Phi(X,Y) := \left\lceil \frac{57X^2}{\eta^2} \left[ \log\Big( \frac{8eX\sqrt{2SA}}{\sqrt{\delta}\eta} \Big) \right]^2 + \frac{24Y}{\eta} \log\Big( \frac{24YSA}{\delta\eta} \Big) \right\rceil.$$

Using a technical lemma (Lem. 14), we can prove that condition (14) holds whenever the number of samples at the pair $(s, a)$ becomes at least equal to

$$\phi_t^{\text{RMODEST}}(s, a) := \Phi(X, Y), \qquad X := \max_{s' \in \mathcal{S}} \sqrt{\widehat{\sigma}_t^2(s'|s, a)}, \qquad Y := 1.$$

We thus execute GOSPRL until there exists a time step $t \geq 1$ such that $b_t(s, a) := \phi_t^{\text{RMODEST}}(s, a)$ samples have been collected at each state-action pair $(s, a) \in \mathcal{S} \times \mathcal{A}$. Although the sampling requirement $b_t$ depends on the time step $t$, this is not an issue from Sect. 3.2 since for any $s \in \mathcal{S}$ and $t \geq 1$, $b_t(s, a)$ is bounded from above due to the fact that $\widehat{\sigma}_t^2(s'|s, a) \leq \frac{1}{4}$. This means that the total requirement for RMODEST is $B_{\text{RMODEST}} = \widetilde{O}(SA/\eta^2)$, which yields the first bound of Lem. 13.

We now turn to the MODEST objective. GOSPRL collects samples until there exists a time step $t$ such that the number of samples at each pair $(s, a)$ is at least equal to

$$\phi_t^{\text{MODEST}}(s, a) := \Phi(X, Y), \qquad X := \sum_{s' \in \mathcal{S}} \sqrt{\widehat{\sigma}_t^2(s'|s, a)}, \qquad Y := S.$$

Introducing $\Gamma(s, a) := \|p(\cdot|s, a)\|_0$ the maximal support of $p(\cdot|s, a)$, we use the following inequality (valid at any time step $t \geq 1$): $\sum_{s' \in \mathcal{S}} \widehat{\sigma}_t(s'|s, a) \leq \sqrt{\Gamma(s, a) - 1}$ (see e.g., 22, Lem. 4). This means that the total requirement for MODEST is $B_{\text{MODEST}} = \widetilde{O}\left(\frac{\sum_{s,a} \Gamma(s,a)}{\eta^2} + \frac{S^2 A}{\eta}\right)$. Plugging in the result of Thm. 1 finally yields the second bound of Lem. 13 (which corresponds to the statement of Lem. 3 in Sect. 4). $\square$

**Lemma 14.** *For any $x \geq 2$ and $a_1, a_2, a_3, a_4 > 0$ such that $a_3 x \leq a_1 \sqrt{x} \log(a_2 x) + a_4 \log(a_2 x)$, the following holds*

$$x \leq \frac{4a_4}{a_3} \log\left(\frac{2a_4 a_2}{a_3}\right) + \frac{128 a_1^2}{9 a_3^2} \left[\log\left(\frac{4a_1 \sqrt{a_2} e}{a_3}\right)\right]^2.$$

*Proof.* Assume that $a_3 x \leq a_1 \sqrt{x} \log(a_2 x) + a_4 \log(a_2 x)$. Then we have $\frac{a_3}{2} x \leq -\frac{a_3}{2} x + a_1 \sqrt{x} \log(a_2 x) + a_4 \log(a_2 x)$. From Lem. 15 we have

$$-\frac{a_3}{2} x + a_1 \sqrt{x} \log(a_2 x) \leq \underbrace{\frac{32 a_1^2}{9 a_3} \left[\log\left(\frac{4a_1 \sqrt{a_2} e}{a_3}\right)\right]^2}_{:=a_0}.$$

Thus we have $x \leq \frac{2a_4}{a_3} \log(a_2 x) + \frac{2a_0}{a_3}$ and we conclude the proof using Lem. 16. $\square$

**Lemma 15** (32, Lem. 8). *For any $x \geq 2$ and $a_1, a_2, a_3 > 0$, the following holds*

$$-a_3 x + a_1 \sqrt{x} \log(a_2 x) \leq \frac{16 a_1^2}{9 a_3} \left[\log\left(\frac{2a_1 \sqrt{a_2} e}{a_3}\right)\right]^2.$$

**Lemma 16.** *Let $b_1$, $b_2$ and $b_3$ be three positive constants such that $\log(b_1 b_2) \geq 1$. Then any $x > 0$ satisfying $x \leq b_1 \log(b_2 x) + b_3$ also satisfies $x \leq 2b_1 \log(2b_1 b_2) + 2b_3$.*

*Proof.* Assume that $x \leq b_1 \log(b_2 x) + b_3$ and set $y = x - b_3$. If $y \leq b_3$, then we have $x \leq 2b_3$. Otherwise, we can write $y \leq b_1 \log(b_2 y + b_2 b_3) \leq b_1 \log(2b_2 y)$. From Lem. 17 we have $y \leq 2b_1 \log(2b_1 b_2)$, which concludes the proof. $\square$

**Lemma 17** (32, Lem. 9). *Let $b_1$ and $b_2$ be two positive constants such that $\log(b_1 b_2) \geq 1$. Then any $x > 0$ satisfying $x \leq b_1 \log(b_2 x)$ also satisfies $x \leq 2b_1 \log(b_1 b_2)$.*

# G Application: Sparse Reward Discovery (TREASURE Problem)

In this section, we focus on the canonical sampling requirement of the TREASURE problem of Sect. 4.1, where each state-action pair must be visited at least once. We illustrate how direct adaptations of existing algorithms are not able to match the guarantees of GOSPRL in Lem. 2.

**Discussion on finite-horizon or discounted PAC-MDP algorithms.** At first glance, an approach to tackle the TREASURE problem could be to consider a well-known PAC exploration algorithm such as RMAX [13] (the same discussion holds for E3 of [34]). In particular, we can examine the ZERORMAX variant proposed in [30]. Indeed the demarcation between known states and unknown states is an algorithmic principle related to the problem at hand: a state is considered known when the number of times each action has been executed at that state is at least $m$ for a suitably chosen $m$ and its reward is set to 0, while an unknown state receives a reward of 1. The set of known states captures what has been sufficiently sampled (and the empirical estimate of the transitions is used), while the set of unknown states drives exploration to collect additional samples. The central concept for analyzing the sample complexity of the algorithm is the escape probability (i.e., the probability of visiting the unknown states), which, in the case of $m = 1$, would amount exactly to the probability of collecting a required sample in the TREASURE problem. However, despite the similarities, ZERORMAX (as well as RF-RL-EXPLORE of [30]) are designed in the infinite-horizon discounted setting or the finite-horizon setting. As such, only a finite number of steps is relevant, and the episode lengths (and resulting sample complexity) directly depend on the discount factor $\gamma$ or on the horizon $H$, respectively. Such approach cannot be employed in the setting of communicating MDPs, where there is no known imposed horizon of the problem, and where the agent must interweave the policy planning and policy execution processes by defining algorithmic episodes. As such, despite bearing high-level similarity with GOSPRL at an algorithmic level, such finite-horizon (or discounted) guarantees cannot be translated to sample complexity for the TREASURE problem.

**Leveraging UCRL.** We now analyze UCRL2 [28], an efficient algorithm for reward-dependant exploration in the infinite-horizon undiscounted setting. In order to tackle the TREASURE problem, a first approach could be to consider true rewards of zero everywhere while the uncertainty around the rewards remains, i.e., the algorithm observes as reward $r(s,a) \sim \sqrt{\frac{1}{N^+(s,a)}}$, which corresponds to the usual uncertainty on the rewards [28], with $N(s,a)$ denoting the number of visits of $(s,a)$ so far. The underlying idea is that as the algorithm visits a state-action pair, its observed reward will decrease, thus favoring the visitation of non-sampled state-action pairs. Yet while this algorithm is fairly intuitive, it appears tricky to directly leverage the analysis of UCRL2 to obtain a guarantee on the time the algorithm requires to solve the TREASURE problem. Indeed, the inspection of the tools used in the regret derivation of UCRL2 does not point out to a step in the analysis which explicitly lower bounds state-action visitations.

Another possibility is to design a non-stationary reward signal to feed to UCRL2. Namely, assigning a reward of 1 if the state is under-sampled and 0 otherwise, corresponds to a sensible strategy (note that this reward signal changes according to the behavior of the algorithm). Yet as explained in [50], for any SSP problem with unit costs, the SSP-regret bound that is obtained from the analysis of average-reward techniques (by assigning a reward of 1 at the goal state, and 0 everywhere else) is worse than that obtained from the analysis of SSP goal-oriented techniques. This difference directly translates into a worse performance of UCRL2-based approaches for the TREASURE problem. Indeed, retracing the analysis of [50, App. B], we obtain that $\widetilde{O}(D_s^3 S^2 A)$ time steps are required to collect a sought-after sample when running the algorithm UCRL2B [22] (which is a variant of UCRL2 that constructs confidence intervals based on the empirical Bernstein inequality rather than Hoeffding's inequality and thus yields tighter regret guarantees). Since the analysis renders the re-use of samples difficult, performing this reasoning for each sought-after state to sample yields a total TREASURE sample complexity of $\widetilde{O}\big(\sum_{s \in \mathcal{S}} D_s^3 S^2 A\big)$, which is always worse than the bound in Lem. 2 since $\max_s D_s = D$.

**Leveraging MAXENT.** At first glance, an alternative and natural approach to visit each state-action pair at least once may be to optimize the MAXENT objective over the state-action space, i.e., maximize the entropy function $H$ over the stationary state-action distributions $\lambda \in \Lambda$,

$$H(\lambda) := \sum_{(s,a) \in \mathcal{S} \times \mathcal{A}} -\lambda(s,a) \log(\lambda(s,a)).$$

This objective — over the state space, yet the extension to the state-action space is straightforward — was studied in [26] in the infinite-horizon discounted setting and in [17] in the infinite-horizon undiscounted setting. Following the latter, there exists a learning algorithm such that, with overwhelming

probability,

$$H(\lambda^\star) - H(\widetilde{\lambda}_t) = \widetilde{O}\left(\frac{DS^{1/3}}{t^{1/3}} + \frac{DS\sqrt{A}}{\sqrt{t}}\right),\tag{16}$$

where $\lambda^\star \in \arg\max_{\lambda \in \Lambda} H(\lambda)$ and $\widetilde{\lambda}_t$ is the empirical state-action frequency at time $t$, i.e., $\widetilde{\lambda}_t(s,a) = \frac{N_t(s,a)}{t}$. The TREASURE sample complexity translates into the first time step $t \geq 1$ such that $\widetilde{\lambda}_t(s,a) \geq \frac{1}{t}$ for all $(s,a) \in \mathcal{S} \times \mathcal{A}$. However, the state-action entropy $H$ corresponds to the sum of a function related to each state-action frequency, and maximizing it provides no guarantee on each summand, i.e., on each state-action frequency. Indeed, assume that there exists a time $t$ such that $\widetilde{\lambda}_t(s,a) \geq \frac{1}{t}$ for all $(s,a) \in \mathcal{S} \times \mathcal{A}$. This implies that $H(\widetilde{\lambda}_t) \geq \frac{SA}{t}\log(t)$. However, the regret bound of Eq. 16 cannot be leveraged to show that $t$ must necessarily be small enough. Overall, it seems that directly optimizing MAXENT is unfruitful in guaranteeing the visitation of each state-action pair at least once, and thus in provably enforcing the TREASURE objective.

Instead of maximizing MAXENT, the discussion above encourages us to optimize the "worst-case" summand of the entropy function, by maximizing over $\Lambda$ the following function

$$F(\lambda) := \min_{(s,a)\in\mathcal{S}\times\mathcal{A}} \lambda(s,a).$$

It is straightforward to show that $F$ is concave in $\lambda$ (as the minimum of $S \times A$ concave functions), as well as 1-Lipschitz-continuous w.r.t. the Euclidean norm $\|\cdot\|_2$, i.e.,

$$\forall(\lambda,\lambda') \in \Lambda^2, |F(\lambda) - F(\lambda')| \leq \|\lambda - \lambda'\|_\infty \leq \|\lambda - \lambda'\|_2.$$

However, $F$ is a non-smooth function, therefore the Frank-Wolfe algorithmic design of [26, 17] cannot be leveraged. Instead, we propose to use the mirror descent algorithmic design of [17, Sect. 5] that can handle general concave functions. It guarantees that there exists a constant $\beta > 0$ such that, with overwhelming probability (here we exclude logarithmic terms for ease of exposition)

$$F(\lambda^\star) - F(\widetilde{\lambda}_t) \leq \frac{\beta D}{t^{1/3}} + \frac{\beta DS\sqrt{A}}{\sqrt{t}}.$$

Introduce $\omega^\star := F(\lambda^\star) = \min_{s,a} \lambda^\star(s,a) \in (0, \frac{1}{SA}]$. We then have

$$F(\widetilde{\lambda}_t) \geq \omega^\star - \frac{\beta D}{t^{1/3}} - \frac{\beta DS\sqrt{A}}{\sqrt{t}}.\tag{17}$$

Equipped with Eq. 17, we can easily prove that if

$$t = \Omega\left(\min\left\{\frac{D^2S^2A}{(\omega^\star)^2}, \frac{D^3}{(\omega^\star)^3}\right\}\right),\tag{18}$$

then $F(\widetilde{\lambda}_t) \geq \frac{1}{t}$, which immediately implies that the TREASURE is discovered. This sample complexity result is quite poor compared to Lem. 2. In particular, it depends polynomially on $(\omega^\star)^{-1}$, which cannot be smaller than $SA$.

## H  Application: Goal-Free Cost-Free Exploration in Communicating MDPs

### H.1  Reward-Free Exploration in Finite-Horizon MDPs vs. Cost-Free Exploration in Goal-Conditioned RL

Jin et al. [30] introduced the reward-free framework in the finite-horizon case, which we recall is a special case of a goal-oriented (i.e., SSP) problem where each episode terminates after exactly $H$ steps. The agent receives as input an accuracy level $\varepsilon > 0$, a confidence level $\delta \in (0,1)$, the state and action spaces, and the horizon $H$, while no knowledge is provided about the transition model $p$. The learning process is decomposed into two phases. ① *Exploration phase:* The agent first collects trajectories from the MDP without a pre-specified reward function and returns an estimate of the transition model $\widehat{p}$. ② *Planning phase:* The agent receives an arbitrary reward function and is tasked

with computing an $\varepsilon$-optimal policy with probability at least $1 - \delta$, without any additional interaction with the environment. The objective is to minimize the duration of the exploration phase needed to simultaneously enforce any requested planning guarantee.

In [30] the reward-free exploration problem is studied for any arbitrary MDP, where there may exist states that are difficult or impossible to reach. The core mechanism in their analysis is to partition the states depending on their ease of being reached within $H$ steps. Specifically, they distinguish between *significant* states, that can be sufficiently visited and whose transition probability can thus be accurately estimated, and *insignificant* states that are too difficult to reach within $H$ steps, but therefore have negligible contribution to any reward optimization.

Interestingly, in the goal-conditioned setting this distinction may no longer be meaningful. By way of illustration, consider any fixed horizon $H$ and the toy environment in Fig. 5. Suppose that the objective is to quickly reach state $z$ (i.e., the goal state is $z$, the starting state is $x$ and all costs are equal to 1). Even though state $y$ is *insignificant* within $H$ steps (in the finite-horizon sense of 30, for any positive "significance level"), it is actually crucial in solving the objective, as $z$ can be reached deterministically in 1 step from $y$. Extrapolating this scenario, in the goal-conditioned setting, we may have an effective horizon of $H = +\infty$ for some goals, which implies that the transition model $p$ must be accurately estimated across the *entire* state-action space to ensure that a near-optimal goal-conditioned policy can be computed.

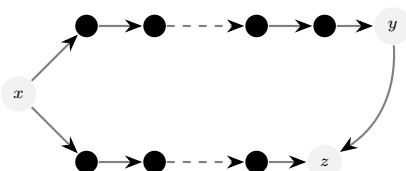

Figure 5: The agent starts at state $x$ and reaches $z$ in $H$ steps with probability $1/2$, and $y$ in $H + 1$ steps with probability $1/2$. From state $y$ the agent deterministically transitions to state $z$ in 1 step.

As a result, the challenges that emerge in the cost-free exploration problem in goal-conditioned RL are orthogonal to the ones in finite-horizon [30]: a *constraint on the environment is added* (all states must now be reachable, Asm. 1), allowing the *removal of the constraint on performance* (which is not limited to $H$ steps anymore) and thus enabling to tackle the more general class of goal-oriented problems.

For a designated goal state $g \in \mathcal{S}$, recall that the SSP objective is to compute a policy $\pi : \mathcal{S} \to \mathcal{A}$ minimizing the cumulative cost before reaching $g$. Formally, the (possibly unbounded) value function is defined as

$$V_\pi(s \to g) := \mathbb{E}\left[ \sum_{t=1}^{\tau_\pi(s \to g)} c(s_t, \pi(s_t)) \,\Big|\, s_1 = s \right],$$

where $\tau_\pi(s \to g) := \inf\{t \geq 0 : s_{t+1} = g \,|\, s_1 = s, \pi\}$ is the (random) number of steps needed to reach $g$ from $s$ when executing policy $\pi$. An optimal policy (if it exists) is denoted by $\pi^\star \in \arg\min_\pi V_\pi(s \to g)$.

Without loss of generality, we consider throughout that the maximum $c_{\max}$ of the cost functions that we intend to consider in the planning phase is equal to 1. On the other hand, the minimum value $c_{\min}$ has a more subtle impact on the type of performance guarantees we can obtain. Following [53], for any cost function $c$ and any pair of initial and goal states $s$ and $g$, we introduce a slack parameter $\theta \in [1, +\infty]$ and we say that a policy $\widehat{\pi}$ is $(\varepsilon, \theta)$-optimal if [14]

$$V^{\widehat{\pi}}(s \to g) \leq \min_{\pi : \mathbb{E}[\tau_\pi(s \to g)] \leq \theta D_{s,g}} V^\pi(s \to g) + \varepsilon. \tag{19}$$

We consider this restricted optimality only in the general cost case of $c_{\min} = 0$, where the $(\varepsilon, +\infty)$-optimal policy may not be proper [50, 45]. In that case, we are interested in finding the best proper policy, which is what the restricted optimality in Eq. 19 enables as it constrains the targeted policy to be proper. This consideration is required in the analysis of [53] when translating the performance from the cost-perturbed MDP to the original MDP, which needs constraining the expected goal-reaching time of the targeted policy.

We are now ready to formally define the *goal-free cost-free exploration* problem. It is characterized by an accuracy level $0 < \varepsilon \leq 1$, a confidence level $\delta \in (0, 1)$, a minimum cost $c_{\min} \in [0, 1]$ and a slack parameter $\theta \in [1, +\infty]$ (and we allow either $c_{\min} = 0$ or $\theta = +\infty$, but not both simultaneously).

---

[14] This reduces to standard $\varepsilon$-optimality for $\theta \to \infty$.

After its exploration phase (whose number of time steps defines the sample complexity of the problem), the agent is expected to be able to compute, with probability at least $1 - \delta$, an $(\varepsilon, \theta)$-optimal goal-conditioned policy $\widehat{\pi}$ for *any* goal state $g \in \mathcal{S}$ and *any* cost function $c \in [c_{\min}, 1]$, i.e., satisfying Eq. 19 for all $s \in \mathcal{S}$.

## H.2 Proof of Lem. 4

We show that instantiating GOSPRL for carefully selected sampling requirements $b_t(s, a)$ enables to obtain the guarantee of Lem. 4. To do so, we build on the sample complexity analysis of solving a fixed-goal SSP problem with a generative model of [53]. Specifically, we introduce the following sampling requirement function

$$\phi(X, y) := \alpha \cdot \left( \frac{X^3 \widehat{\Gamma}}{y \varepsilon^2} \log\left( \frac{XSA}{y \varepsilon \delta} \right) + \frac{X^2 S}{y \varepsilon} \log\left( \frac{XSA}{y \varepsilon \delta} \right) + \frac{X^2 \widehat{\Gamma}}{y^2} \log^2\left( \frac{XSA}{y \delta} \right) \right), \qquad (20)$$

where $\alpha > 0$ is a numerical constant and $\widehat{\Gamma} := \max_{s,a} \|\widehat{p}(\cdot|s, a)\|_0 \leq \Gamma$ is the largest support of $\widehat{p}$.

The sampling requirement function of Eq. 20 instantiated for specific values of $X$ and $y$ is used to guide the GOSPRL algorithm. Specifically, the analysis distinguishes between two cases: *either* $c_{\min} > 0$ and the cost function considered in the planning phase can be the same as the original one, *or* $c_{\min} = 0$ and all costs incur an additive perturbation of $\varepsilon/(\theta D) > 0$ (as considered in the analysis of [53]). As stated in Sect. 4.3, we set $\omega := \max \{c_{\min}, \varepsilon/(\theta D)\}$, which is guaranteed to be positive since we enforce either $c_{\min} = 0$ or $\theta = +\infty$, but not both simultaneously. As such, in Eq. 20 we define $y := \omega$ to be equal to the minimum cost of either the true or the perturbed cost function. As for the value of $X$, we perform the following distinction of cases.

① First let us assume that the learning agent has prior knowledge of the diameter $D$ of the MDP. Then we set $X := D$. Since the analysis of [53] accurately estimates the transition kernel and thus holds for arbitrary cost function in $[\omega, 1]$, we can ensure that collecting at least $\phi(D, \omega)$ samples from each state-action pair provides the $\varepsilon$-optimality cost-free planning guarantee of Lem. 4. The total time required to collect such samples is upper bounded by $DSA\phi(D, \omega^{-1})$, which directly yields the sample complexity guarantee stated in Lem. 4.

② Second we show that we can relax the assumption of knowing the diameter $D$ without altering the sample complexity guarantee. To do so, we begin the algorithm by a procedure which computes a quantity $\widehat{D}$ such that $D \leq \widehat{D} \leq D(1 + \varepsilon)$ with high probability. From App. I.1, this can be done in $\widetilde{O}(D^3 S^2 A/\varepsilon^2)$ time steps by leveraging GOSPRL. We thus begin the algorithm by running such diameter-estimation subroutine. Crucially, we note that its sample complexity is subsumed in the total sample complexity of Lem. 4. Then we simply apply the reasoning in case ① by considering $X := \widehat{D}$ in the allocation of Eq. 20 instead of $X = D$. Since $\widehat{D}$ is a sufficiently tight upper bound on $D$ (i.e., $\widehat{D} = O(D)$), we ultimately obtain the same sample complexity guarantee as in case ①.

# I  Other Applications

In this section, we provide additional applications where GOSPRL can be leveraged to readily obtain an online learning algorithm. We first summarize them here.

**Diameter estimation (see App. I.1).** GOSPRL can be leveraged to estimate the MDP diameter $D$. In App. I.1 we develop a GOSPRL-based procedure that computes an estimate $\widehat{D}$ such that $D \leq \widehat{D} \leq (1 + \varepsilon)D$ in $\widetilde{O}(D^3 S^2 A/\varepsilon^2)$ time steps. This improves on the diameter estimation procedure recently devised in [65] by a multiplicative factor of $DS^2$. As $\widehat{D}$ provides an upper bound on the optimal bias span $sp(h^\star)$, our procedure may be of independent interest for initializing average-reward regret-minimization algorithms that leverage prior knowledge of $sp(h^\star)$ (as done in e.g., [65]).

**PAC-policy learning (see App. I.2).** One of the most common $\mathcal{SO}$-based settings is the computation of an $\varepsilon$-optimal policy via sample-based value iteration. Since GOSPRL is agnostic to how the sampling requirements are generated, we can easily integrate it with any state-of-the-art $\mathcal{SO}$-based algorithm and directly inherit its properties. For instance, in App. I.2 we show that GOSPRL can

---

**Algorithm 2** GOSPRL-based procedure to estimate the diameter

---

1: **Input:** accuracy $\varepsilon > 0$, confidence level $\delta \in (0, 1)$.
2: Set $W := \frac{1}{2}$ and $\|\widetilde{v}\|_\infty^\infty := 1$.
3: **while** $\|\widetilde{v}\|_\infty^\infty > W$ **do**
4:     Set $W \leftarrow 2W$.
5:     Set the accuracy $\eta := \frac{\varepsilon}{W}$.
6:     Collect additional samples by running GOSPRL for the MODEST problem with accuracy $\frac{\eta}{2}$ and confidence level $\delta$.
7:     **for** each state $s \in \mathcal{S}$ **do**
8:         Compute a vector $\widetilde{v}(\cdot \rightarrow s)$ using EVI for SSP, with goal state $s$, unit costs and VI precision $\mu_{\mathrm{VI}} := \frac{\min\{1, \varepsilon\}}{2}$ (see App. A).
9:     **end for**
10: **end while**
11: **Output:** the quantity $\widehat{D} := (1 + 2\eta\|\widetilde{v}\|_\infty^\infty)\|\widetilde{v}\|_\infty^\infty$.

---

be easily combined with BESPOKE [62] to obtain a competitive online learning algorithm for the policy learning problem. In fact, the sample complexity of the resulting algorithm is only a factor $D$ worse than existing online learning algorithms in the worst case and, leveraging the refined problem-dependent bounds of BESPOKE, it is likely to be superior in many MDPs.

**Bridging bandits and MDPs with GOSPRL (see App. I.3).** In multi-armed bandit (MAB) an agent directly collects samples by pulling arms. If we map each arm to a state-action pair, we can see any MAB algorithm as having access to an $\mathcal{SO}$. As such, we can readily turn any bandit algorithm into an RL online linear algorithm by calling GOSPRL to generate the samples needed by the MAB algorithm. Exploiting this procedure, in App. I.3 we show how we can tackle problems such as *best-state identification* and *active exploration* (i.e., state-signal estimation) in the communicating MDP setting, for which no specific online learning algorithm exists yet.

## I.1 Application: Diameter Estimation

GOSPRL can be leveraged to estimate the diameter $D$ which is a quantity of interest in the average-reward setting. Indeed, $D$ dictates the performance of reward-based no-regret algorithms [28], and some works assume that an upper bound on the optimal bias span $sp(h^\star)$ is known (e.g., [44]). Since we have $sp(h^\star) \leq r_{\max} D$ (e.g., [8]), upper bounding $D$ enables to relax this assumption. Recently, for such purpose of upper bounding $sp(h^\star)$, [65] developed an initial procedure based on successive applications of UCRL2 that can compute an estimate $\widehat{D}$ such that $D \leq \widehat{D} \leq (1 + \varepsilon)D$ in $\widetilde{O}(D^4 S^4 A/\varepsilon^2)$ time steps (see [65, App. D & Alg. 3 "LD: Learn the Diameter"]). In Alg. 2 we derive an iterative estimation procedure based on GOSPRL which can compute such upper bound of $D$ faster, namely in $\widetilde{O}(D^3 S^2 A/\varepsilon^2)$ time steps, while simultaneously providing an accurate estimation of the transition dynamics. As such it may be an initial procedure of independent interest for regret-minimization algorithms in the average-reward setting. Note that the procedure is similar to the one considered in [53] to estimate an upper bound of the SSP-diameter of a given SSP problem in the generative model case, while here we focus on the estimation of the diameter (i.e., worst-case SSP diameter) in the *online* case by leveraging GOSPRL.

We define a notation used throughout the section, $\|U\|_\infty^\infty := \max_{s, s'} U(s \rightarrow s')$, which holds for any quantity $U$ that can be naturally mapped to a $\mathcal{S} \times \mathcal{S}$ matrix.

**Lemma 18.** *With probability at least $1 - \delta$, Alg. 2:*

- *has a sample complexity bounded by $\widetilde{O}(D^3 S^2 A/\varepsilon^2)$,*
- *requires at most $\log_2(D(1 + \varepsilon)) + 1$ inner iterations,*
- *solves the MODEST problem for an accuracy level $\eta > 0$ and outputs an optimistic $\mathcal{S} \times \mathcal{S}$ matrix $\widetilde{v}$ s.t. $\frac{\varepsilon}{2D} \leq \eta \leq \frac{\varepsilon}{\|\widetilde{v}\|_\infty^\infty}$,*
- *outputs a quantity $\widehat{D} := (1 + 2\eta\|\widetilde{v}\|_\infty^\infty)\|\widetilde{v}\|_\infty^\infty$ that verifies $D \leq \widehat{D} \leq (1 + 2\varepsilon(1 + \varepsilon))(1 + \varepsilon)D$.*

*Proof.* We will assume throughout that the event $\mathcal{E}$ (defined in App. A) holds. We now give a useful statement stemming from optimism:

"At any stage of Alg. 2, for any given goal state, denote by $\widetilde{v}$ the vector computed using EVI for SSP. Then under the event $\mathcal{E}$, we have component-wise (i.e., starting from any non-goal state): $\widetilde{v} \leq \min_\pi V_p^\pi \leq D$."

To prove this useful statement, we observe that the first inequality stems from Lem. 5 of App. A while the second inequality uses the definition of the diameter $D$ and the fact that the considered costs are equal to 1.

Now, denote by $n$ the iteration index of the Alg. 2 (starting at $n = 1$), so that $W_n = 2^n$. Introduce $N := \min\{n \geq 1 : \|\widetilde{v}_n\|_\infty^\infty \leq W_n\}$. We have $\|\widetilde{v}_n\|_\infty^\infty \leq D$ at any iteration $n \geq 1$ from the useful statement on optimism above. Since $(W_n)_{n \geq 1}$ is a strictly increasing sequence, Alg. 2 is bound to end in a finite number of iterations (i.e., $N < +\infty$), and given that $W_{N-1} \leq \|\widetilde{v}_{N-1}\|_\infty^\infty \leq D$, we get $N \leq \log_2(D) + 1$. Moreover, we have $\|\widetilde{v}_N\|_\infty^\infty \leq W_N$ and $\eta_N = \frac{\varepsilon}{W_N}$, which implies that $\eta_N \leq \frac{\varepsilon}{\|\widetilde{v}_N\|_\infty^\infty}$. Moreover, combining $W_{N-1} \leq D$ and $W_{N-1} = \frac{W_N}{2} = \frac{\varepsilon}{2\eta_N}$ yields that $\frac{\varepsilon}{2D} \leq \eta_N$.

Denote by $\eta := \eta_N$ the achieved MODEST accuracy at the end of Alg. 2. Plugging in the guarantee of Prop. 3 yields a sample complexity of

$$\widetilde{O}\Big(\frac{DS^2 A}{\eta^2}\Big) = \widetilde{O}\Big(\frac{D^3 S^2 A}{\varepsilon^2}\Big).$$

Denote by $\widetilde{v} := \widetilde{v}_N$ the optimistic matrix output by Alg. 2. Consider $(s_1, s_2) \in \arg\max_{(s,s')} \min_\pi \mathbb{E}[\tau_\pi(s \to s')]$. Denote by $\widetilde{\pi}$ the greedy policy w.r.t. the vector $\widetilde{v}(\cdot \to s_2)$ in the optimistic model with goal state $s_2$. Then we have

$$D = \min_\pi \mathbb{E}[\tau_\pi(s_1 \to s_2)]\mathbb{E}[\tau_{\widetilde{\pi}}(s_1 \to s_2)] \overset{(a)}{\leq} (1 + 2\eta\|\mathbb{E}[\widetilde{\tau}_{\widetilde{\pi}}]\|_\infty^\infty)\mathbb{E}[\widetilde{\tau}_{\widetilde{\pi}}(s_1 \to s_2)]$$

$$\overset{(b)}{\leq} (1 + 2\eta(1+\varepsilon)\|\widetilde{v}\|_\infty^\infty)(1+\varepsilon)\widetilde{v}(s_1 \to s_2) \leq (1 + 2\eta(1+\varepsilon)\|\widetilde{v}\|_\infty^\infty)(1+\varepsilon)\|\widetilde{v}\|_\infty^\infty := \widehat{D}$$

$$\overset{(c)}{\leq} (1 + 2\eta(1+\varepsilon)\|\widetilde{v}\|_\infty^\infty)(1+\varepsilon)D \overset{(d)}{\leq} (1 + 2\varepsilon(1+\varepsilon))(1+\varepsilon)D,$$

where (a) corresponds to an SSP simulation lemma argument (see [45, Lem. B.4]; [52, Lem. 3]) given that a MODEST accuracy of $\eta$ is fulfilled, (b) comes from the value iteration precision $\mu_{\text{VI}} := \frac{\min\{1,\varepsilon\}}{2}$ which implies that $\mathbb{E}[\widetilde{\tau}_{\widetilde{\pi}}] \leq (1 + 2\mu_{\text{VI}})\widetilde{v} \leq (1+\varepsilon)\widetilde{v}$ component-wise according to Lem. 5 of App. A, (c) is implied by the useful statement on optimism given at the beginning of the proof, and finally (d) leverages that $\eta\|\widetilde{v}\|_\infty^\infty \leq \varepsilon$. $\qquad\square$

## I.2 Application: PAC-Policy Learning

One of the most common $\mathcal{SO}$-based settings is the computation of an $\varepsilon$-optimal policy via sample-based value iteration. Since GOSPRL is agnostic to how the sampling requirements are generated, we can easily integrate it with any state-of-the-art $\mathcal{SO}$-based algorithm and directly inherit its properties. For instance, consider the BESPOKE algorithm introduced by [62]. BESPOKE proceeds through phases and at the beginning of each phase $k$, it determines the additional number of samples $n_{sa}^{k+1}$ that need to be generated at each state-action pair $(s, a)$ based on the estimates of the model and reward of the MDP computed so far. Then it simply queries the $\mathcal{SO}$ as needed and it moves to the following phase. In order to turn BESPOKE into an online learning algorithm, we can simply replace the query step by running GOSPRL until $n_{sa}^{k+1}$ samples are generated and then move to the next phase. Furthermore, let $b(s, a)$ be the total number of samples required by BESPOKE in each state-action pair as stated by [62, Thm. 2], then we can directly apply Thm. 1 and obtain the sample complexity of the online version of BESPOKE (ONLINE-BESPOKE). As discussed in Sect. 3 the resulting complexity is *at most* a factor $D$ larger than the one of (offline) BESPOKE plus an additional term of order $\widetilde{O}(D^{3/2}S^2 A)$ independent from the desired accuracy $\varepsilon$. It is interesting to contrast this result with existing online algorithms for this problem. While to the best of our knowledge, there is no algorithm specifically designed for optimal policy learning, we can rely on regret-to-PAC conversion (see e.g., [29, Sect. 3.1]) to derive sample complexity guarantees for existing regret minimization algorithms and do a qualitative comparison.[15] For instance, we can use EULER [61] to derive an

---

[15]Regret minimization guarantees are usually provided for the finite-horizon setting, while BESPOKE is designed for the discounted setting. Furthermore, the $\varepsilon$-optimality guarantees for $\mathcal{SO}$-based algorithms are typically defined in $\ell_\infty$ norm, while the regret-to-PAC conversion only provides guarantees on average w.r.t. the initial distribution.

$\varepsilon$-optimal policy. If we consider a worst-case analysis, EULER achieves the same sample complexity of BESPOKE, which in turn matches the lower bound of [5]. As a result, ONLINE-BESPOKE would be a factor $D$ suboptimal w.r.t. to EULER. Nonetheless, our $\mathcal{SO}$-to-online learning conversion approach enables ONLINE-BESPOKE to directly benefit from the problem-dependent performance of BESPOKE, which in many MDPs may outperform the guarantees obtained by using EULER as a online learning algorithm for policy optimization.

### I.3    Application: Bandit Problems with MDP Dynamics

#### I.3.1    Algorithmic protocol

The sampling procedure GOSPRL provides an effective way to collect samples for states of the agent's choosing, and can thus be related to the multi-armed bandit setting by mapping arms (in bandits) to states (in MDPs). From Thm. 1, each state can now be "pulled" within $\widetilde{O}(D)$ time steps (instead of a single time step in the bandit case). This allows to naturally extend some *pure exploration* problems from the bandit setting to the communicating MDP setting. The algorithmic protocol alternates between the two following strategies:

1. the "bandit algorithm" identifies the arm(s), i.e., state(s), from which a sample is desired,
2. GOSPRL is executed to collect a sought-after sample as fast as possible.

To illustrate our decoupled approach we consider the two following problems: best-state identification (App. I.3.2) and reward-estimation, a.k.a. active exploration (App. I.3.3).

#### I.3.2    Best-state identification

This is the MDP extension of the best-arm identification problem in bandits [3]. Each state $s \in \mathcal{S} := \{1, \ldots, S\}$ is characterized by a reward function $r_s$. For the sake of simplicity, we assume that the rewards are in $[0, 1]$ and that there is a unique highest-rewarding state $s^\star := \arg \max_s r_s$. Let $r^\star := r_{s^\star}$. Consider a budget of $n$ steps. The objective is to bound the probability of error $e_n := \mathbb{P}(J_n \neq s^\star)$, where $J_n$ is the state from which we desire a sample at step $n$. For $s \neq s^\star$, we introduce the following suboptimality measure of state $s$: $\Delta_s := r^\star - r_s$. We introduce the notation $(i) \in \{1, \ldots, S\}$ to denote the $i$–th best arm (with ties break arbitrarily). The hardness of the task will be characterized by the following quantities $H_1 := \sum_{s \in \mathcal{S}} \frac{1}{\Delta_s^2}$ and $H_2 := \max_{s \in \mathcal{S}} s \Delta_{(s)}^{-2}$. These quantities are equivalent up to a logarithmic factor since we have $H_2 \leq H_1 \leq \log(2S)H_2$. A fully connected MDP with known and deterministic transitions amounts to a multi-armed bandit problem of $K := S$ arms for our problem, thus the SUCCESSIVE REJECTS algorithm [3] directly yields the following bound after $j$ time steps

$$e_j \leq \frac{S(S-1)}{2} \exp\left(-\frac{j - SA}{\overline{\log}(S)H_2}\right), \quad \text{where } \overline{\log}(S) := \frac{1}{2} + \sum_{i=1}^{S} \frac{1}{i}.$$

In a general MDP, we combine GOSPRL (for the sample collection) with the SUCCESSIVE REJECTS algorithm (for deciding which sample to collect). Consider any large enough budget of $n = \Omega(D^{3/2}S^2A)$ time steps. Denote by $j_n$ the number of time steps during which GOSPRL effectively collects the desired sample stipulated by the SUCCESSIVE REJECTS algorithm. Thm. 1 yields that $n = \widetilde{O}\big(Dj_n + D^{3/2}S^2A\big)$, which means that $j_n = \widetilde{\Omega}\Big(\frac{n - D^{3/2}S^2A}{D}\Big)$. Therefore we obtain the following guarantee.

**Lemma 19.** *In any unknown communicating MDP with unique highest-rewarding state $s^\star$, combining* GOSPRL *with the* SUCCESSIVE REJECTS *algorithm [3] yields the existence of a polynomial function $p$ such that the probability $e_n$ of wrongly identifying the "best state" $s^\star$ at time step $n$ is upper bounded by*

$$e_n \leq p(S, A, D, n) \exp\left(-\frac{n - D^{3/2}S^2A}{D \log(S)H_2}\right),$$

*which corresponds to an exponential decrease w.r.t. $n$ whenever $n$ is large enough (i.e., after the $D^{3/2}S^2A$ burn-in phase).*

### I.3.3 Reward estimation (a.k.a. active exploration)

The objective of this problem in bandits (resp. MDPs) is to accurately estimate the mean pay-off (resp. the average reward signal) at each arm (resp. state). Note that this problem was originally studied in the bandit setting (see e.g., [14]) and recently extended in ergodic MDPs by [51] using a Frank-Wolfe approach. The extension to communicating MDPs remained an open question, and it becomes immediately addressed with GOSPRL. We recall the problem formulation: for a desired accuracy $\varepsilon > 0$, for each state-action pair $(s, a) \in \mathcal{S} \times \mathcal{A}$ with mean reward $r_{s,a}$ in $[0, 1]$, we seek to output an estimate $\widehat{r}_{s,a}$ such that $|\widehat{r}_{s,a} - r_{s,a}| \leq \varepsilon$. Under the GOSPRL framework, it is sufficient to visit each state-action pair at least $\Omega(\varepsilon^{-2})$ times, which directly induces the following sample complexity guarantee.

**Lemma 20.** *In any unknown communicating MDP, GOSPRL can reach any reward-estimation accuracy $\varepsilon > 0$ with high probability under a sample complexity scaling as*

$$\widetilde{O}\left( \frac{DSA}{\varepsilon^2} + D^{3/2} S^2 A \right).$$

### I.3.4 Comments

**Distinction between regret and sample complexity.** Note that the results above (Lem. 19 and 20) do not provide any guarantee on the *regret* of the corresponding algorithms (which is often the metric of interest in sequential learning). Indeed, our algorithmic approach does not track nor adapt to a notion of optimal performance. Likewise, there remains to derive lower bounds on these problems extended to MDPs, in order to quantity the optimality of our procedure. Nonetheless, our decoupled approach is, to the best of our knowledge, the first method with provably bounded sample complexity that can successfully extend classical bandit problems (such as the two aforementioned ones) to communicating MDPs.

**On the link between MDPs and bandits with a special form of transportation costs.** Under the mapping between bandit arms and MDP states, our sampling paradigm has the effect of casting any MDP as a bandit problem with *transportation costs* between arms. In our setting, the transportation cost from a state to another is unknown, initially unbounded and has to be refined over the learning process (the asymptotically optimal cost amounts to the shortest path distance between the two states). We believe that such a setting of unknown and learnable transportation costs is an interesting formalism to study in the bandit setting, as it may then be applied to the MDP extension and allow for smart algorithms that take into account each transportation cost when proposing the arm/state from which a sample is desired (i.e., in part 1 of the algorithmic protocol given at the beginning of App. I.3). For completeness, it is worth mentioning that some papers study various settings of movement/switching costs between arms (see e.g., 19, 36), yet none of these settings can be leveraged for our problem.

## J   On Ergodicity

In this section we explain why the ergodic setting (Asm. 3) and the more general communicating setting of Asm. 1 effectively set the boundary on the difficulty of the problem, in the sense that in an ergodic MDP any sampling requirement is eventually fulfilled, whatever the policy executed.

**Assumption 3** ($M$ is ergodic). *For any stationary policy $\pi$, the corresponding Markov chain $P_\pi$ is ergodic, i.e., all states are aperiodic and recurrent.*

Consider any sampling requirements $b : \mathcal{S} \to \mathbb{N}$ and fix any stationary policy $\pi$. It induces an ergodic chain $P_\pi$ with stationary distribution denoted by $\mu_\pi \in \Delta(S)$. Let $\mu_{\pi,\min} := \min_{s \in \mathcal{S}} \mu_\pi(s) > 0$. We assume without loss of generality that $P_\pi$ is reversible with spectral gap $\gamma_\pi > 0$. (Otherwise, in the non-reversible case, the dependency on $\gamma_\pi$ in Eq. 21 and thus in Eq. 22 is simply replaced by the pseudo-spectral gap introduced in [41].) It is well-known (see e.g., [27, 41]) that with probability at least $1 - \delta$, for any $s \in \mathcal{S}$ and $t \geq 1$,

$$\left| \frac{N_{\pi,t}(s)}{t} - \mu_\pi(s) \right| \leq \sqrt{\frac{2 \log\left( \frac{S}{\delta} \sqrt{\frac{2}{\mu_{\pi,\min}}} \right)}{\gamma_\pi t}} + \frac{20 \log\left( \frac{S}{\delta} \sqrt{\frac{2}{\mu_{\pi,\min}}} \right)}{\gamma_\pi t}, \tag{21}$$

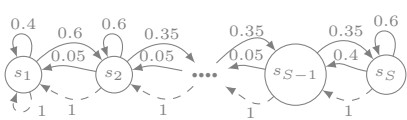

(a) Reward-free RiverSwim ($S = 6$ states)

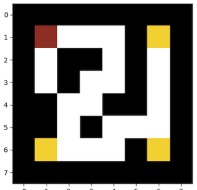

(b) Corridor gridworld ($S = 24$ states)

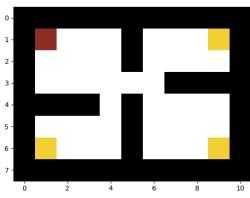

(c) 4-room gridworld ($S = 43$ states)

Figure 6: The three domains considered in Fig. 1. For the gridworlds (b) and (c), the red tile is the starting state, yellow tiles are terminal states that reset to the starting state, and black tiles are reflecting walls (see §"Details on environments").

which implies that

$$
N_{\pi,t}(s) \geq t\mu_{\pi,\min} - \sqrt{t}\sqrt{\frac{2\log\left(\frac{S}{\delta}\sqrt{\frac{2}{\mu_{\pi,\min}}}\right)}{\gamma_\pi}} - \frac{20\log\left(\frac{S}{\delta}\sqrt{\frac{2}{\mu_{\pi,\min}}}\right)}{\gamma_\pi}.
$$

In particular, we can guarantee that $N_{\pi,t}(s) \geq b(s)$ for any $s \in \mathcal{S}$ whenever

$$
t = \Omega\left(\frac{\max_{s \in \mathcal{S}} b(s)}{\gamma_\pi \mu_{\pi,\min}} + \frac{1}{\gamma_\pi \mu_{\pi,\min}^2}\right). \tag{22}
$$

This shows that any policy inevitably meets the sampling requirements in the ergodic setting. Moreover we see that in the case of sampling requirements $b$ that are evened out across the state space, better performance should be achieved by policies with more uniform stationary distributions (i.e., $\mu_{\pi,\min} \gg 0$) and with good mixing properties (i.e., $\gamma_\pi \gg 0$).

## K   Experiments

This section complements the experiments reported in Sect. 5. We provide details about the algorithmic configurations and the environments as well as additional experiments.

**Details on Fig. 1 and Fig. 7.** Fig. 1 reports, as a function of time $t$, the proportion $\mathcal{P}_t$ of states that at time $t$ satisfy the sampling requirements of the TREASURE-10 problem (i.e., $b(s, a) = 10$). Formally, $\mathcal{P}_t := |\{s \in \mathcal{S} : \forall a \in \mathcal{A}, N_t(s, a) \geq b(s, a)\}| \cdot S^{-1}$. As such, all sampling requirements are met as soon as $\mathcal{P}_t = 1$, meaning that the black line $y = 1$ on the y-axis characterizes our objective. Furthermore, we report in Fig. 7 results on additional domains (see below).

**Details on environments.** The three domains considered in Fig. 1 are given in Fig. 6. The first one corresponds to a reward-free version of the RiverSwim domain introduced in [48], which is a stochastic chain with 6 states and 2 actions classically used for testing exploration algorithms. The other two domains are gridworlds. In Fig. 7 we test on a larger RiverSwim domain with 36 states and three additional gridworlds that are given in Fig. 8. Throughout our experiments, the gridworld domains are defined as follows. The agent can move using the cardinal actions (Right, Down, Left, Up). An action fails with probability $p_f = 0.1$, in which case the agent follows (uniformly) one of the other directions. The starting state is shown in red. Yellow tiles are terminal states that, when reached, deterministically reset to the starting state. The black walls act as reflectors, i.e., if the action leads against the wall, the agent stays in the current position with probability 1. The gridworlds are all reward-free, except the one in Fig. 8a where the blue tile incurs large negative environmental reward: it is thus a *trap state* which should be avoided as much as possible. Finally, in the experiments with the randomly generated Garnet environments and state-action requirements (Fig. 2), we guarantee the MDPs randomly generated to be communicating by setting $p(s_0|s, a) \geq 0.001$ for every $(s, a)$ and an arbitrary state $s_0$.

**Algorithmic details.** For all experiments and all considered algorithms, we choose a scaling factor $\alpha_p = 0.1$ of the confidence intervals of the transition probabilities (which enables to speed up the learning, see e.g., [23]), as well as a confidence level set to $\delta = 0.1$. Recall that for GOSPRL, in

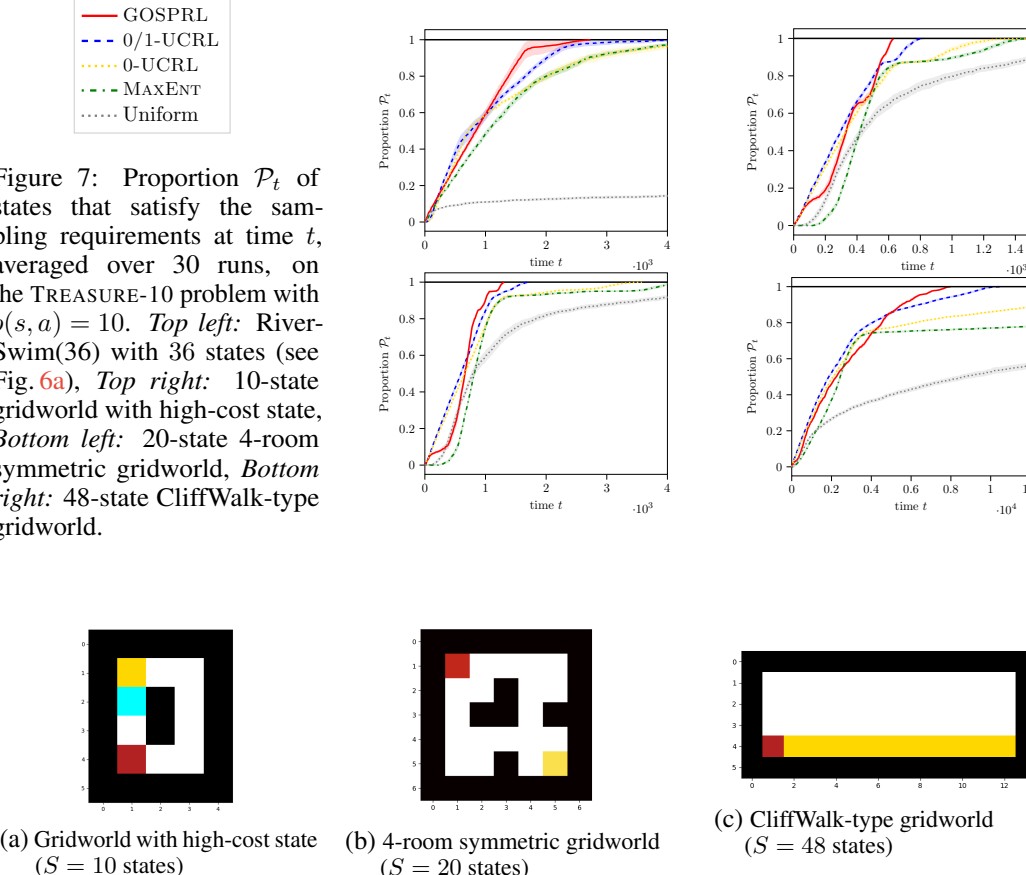

Figure 7: Proportion $\mathcal{P}_t$ of states that satisfy the sampling requirements at time $t$, averaged over 30 runs, on the TREASURE-10 problem with $b(s,a) = 10$. *Top left:* River-Swim(36) with 36 states (see Fig. 6a), *Top right:* 10-state gridworld with high-cost state, *Bottom left:* 20-state 4-room symmetric gridworld, *Bottom right:* 48-state CliffWalk-type gridworld.

(a) Gridworld with high-cost state ($S = 10$ states)

(b) 4-room symmetric gridworld ($S = 20$ states)

(c) CliffWalk-type gridworld ($S = 48$ states)

Figure 8: The three gridworlds considered in Fig. 7. The blue tile in (a) is a "trap state" that incurs large negative environmental reward and should thus be avoided as much as possible.

the case of state-only requirements, a state $s$ is considered as under-sampled and is thus a goal state if $\sum_{a \in \mathcal{A}} N(s,a) < b(s)$, while in the case of state-action requirements, a state $s$ is considered as under-sampled if $\exists a \in \mathcal{A}, N(s,a) < b(s,a)$. We consider the following initial phase for GOSPRL (i.e., when all states are under-sampled): we select as goal states those minimizing the "remaining budget" $b(s) - N(s)$ for state-only requirements (or $\sum_{a \in \mathcal{A}} \max\{b(s,a) - N(s,a), 0\}$ for state-action requirements), which has the effect of shortening the length of the initial phase. In the case of state-action requirements, once a sought-after goal state $s$ is reached, GOSPRL selects an under-sampled action $a$ whose gap $b(s,a) - N(s,a)$ is maximized. We note that this design choice can be observed in Fig. 1 and 7 where GOSPRL seeks to "even out" its sampling strategy, with a steady increase in $(\mathcal{P}_t)$, instead of exhausting the requirements state after state.

**GOSPRL-for-MODEST algorithm.** Here we detail the GOSPRL-for-MODEST algorithm used in the MODEST experiment of Fig. 3. The GOSPRL sampling requirements are computed using a decreasing MODEST accuracy $\eta$, which enables the algorithm to be accuracy-agnostic like the WEIGHTEDMAXENT heuristic to which it is compared. GOSPRL-for-MODEST starts at an initial accuracy of $\eta \leftarrow 1$ and iteratively performs the two following steps until the algorithm ends: *i)* it requires a sampling requirement of $b_t^{\text{MODEST}}(s,a) = \alpha_b \Phi\big(\sum_{s' \in \mathcal{S}} \sqrt{\widehat{\sigma}_t^2(s'|s,a)}, S\big)$, where $\Phi$ is defined after Eq. 3 for accuracy $\eta$ and where $\alpha_b = 0.01$ is a scaling factor to speed up the learning; and *ii)* when the sampling requirements are fulfilled by GOSPRL, it sets $\eta \leftarrow \eta/2$ and goes back to the first step.

**Dependencies.** For each environment of Fig. 6 on the TREASURE-10 problem (i.e., $b(s,a) = 10$, $B = 10SA$), we compute in Tab. 1 the sample complexity of GOSPRL run with known dynamics, to put aside the learning component so that its corresponding sample complexity can be bounded exactly by $BD$ or by $\sum_s b(s)D_s$ according to the analysis in Sect. 3.1. Both bounds are reported in Tab. 1: we observe that the second (more state-dependent) quantity is tighter and more preferable than the

Table 1: For the TREASURE-10 problem, we report the quantities $BD$, $\sum_s b(s)D_s$ and the sample complexity of GOSPRL run with known dynamics (averaged over 30 runs), on the 3 domains of Fig. 6.

| *Environment* | $BD$ | $\sum_s b(s)D_s$ | Sample comp. of GOSPRL run with known dynamics $p$ |
|---|---|---|---|
| RiverSwim(6) | 1766.7 | 958.7 | 249.9 |
| Corridor gridworld(24) | 24375.6 | 13695.2 | 3156.5 |
| 4-room gridworld(43) | 27399.7 | 19048.3 | 3342.5 |

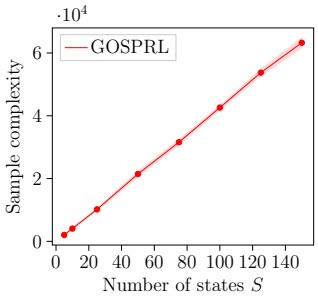

Figure 9: Sample complexity of GOSPRL in randomly generated Garnet MDPs **for increasing values of** $S$, with all other parameters fixed ($A$, $\beta$, $\overline{U}$) as in Fig. 2. Results are averaged over 5 Garnets, each for 12 runs.

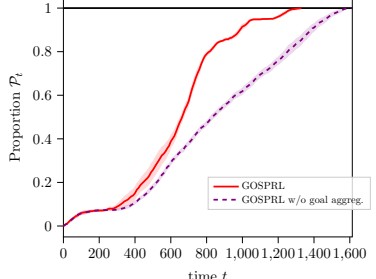

Figure 10: **Impact of goal aggregation on GOSPRL.** Proportion $\mathcal{P}_t$ averaged over 30 runs, on the TREASURE-10 problem with $b(s,a) = 10$ on the environment of Fig. 8b.

first. Despite both bounds being loose w.r.t. the actual algorithmic performance, they can effectively capture the difficulty of the problem (in a relative sense where the higher the bounds, the higher the sample complexity). We also recall from Sect. 5 that there exist simple worst-case problems (see e.g., Fig. 4) where these bounds are tight, i.e., where the sample complexity of GOSPRL (whether the dynamics are known or not) must directly scale with these diameter quantities. Notice that running GOSPRL with known dynamics corresponds to deploying an optimal *greedy* strategy (i.e., by minimizing each time to reach under-sampled states in a sequential fashion), which is likely not the optimal non-stationary solution (which would involve solving a sort of highly difficult, online travelling salesman problem), see App. C.3 for additional discussion. Finally, we study the sample complexity of GOSPRL across similar MDPs with increasing number of states to see how that dependence pans out. Fig. 9 reports the sample complexity of GOSPRL in randomly generated Garnet MDPs for increasing values of $S$. We observe that as expected, the sample complexity scales linearly with $S$.

**Impact of goal aggregation on GOSPRL.** GOSPRL iteratively aggregates the undersampled states into a *meta-goal* for which it computes an optimistic goal-oriented policy. While it is possible to focus on specific goal states as mentioned in App. B without affecting the sample complexity guarantee, performing the goal aggregation leads to shorter and more successful sample collection attempts. We observe in Fig. 10 that this indeed translates into better empirical performance. Indeed, GOSPRL collects the prescribed samples faster than a version of GOSPRL that selects uniformly at random a single goal state among all undersampled states (i.e., that does not perform goal state aggregation).

**Impact of cost shaping on GOSPRL.** While GOSPRL in Alg. 1 considers unit costs for each SSP problem it constructs, any non-unit costs can be designed as long as they are positive and bounded. In particular, detering costs may be assigned to trap states with large negative environmental reward that the agent seeks to avoid. To study this, we consider the gridworld of Fig. 8a where the blue tile is a trap state that the agent must avoid as much as possible. For ease of exposition we consider here sampling requirements concentrated at the terminal state in yellow denoted by $y \in \mathcal{S}$, i.e., $b(y,a) = 10$ for any $a \in \mathcal{A}$. We compare GOSPRL with a cost-weighted GOSPRL where a cost of 10 is set at the blue trap state during each SSP planning step. Tab. 2 shows that while the sample complexity of cost-weighted GOSPRL is worsened, the number of visits to the undesirable trap state is considerably decreased w.r.t. GOSPRL. This makes sense since the shortest path from the red starting state to the sought-after yellow terminal state goes through the blue trap state, so a trade-off

Table 2: **Impact of cost shaping on GOSPRL.** Environment of Fig. 8a. Sampling requirement are concentrated at the yellow terminal state $y \in \mathcal{S}$, i.e., $b(y, a) = 10$ for all $a \in \mathcal{A}$. Cost-weighted GOSPRL sets a cost of 10 (instead of 1) at the blue trap state during each SSP planning step. Values are averaged over 30 runs.

|  | GOSPRL (Alg. 1) | Cost-weighted GOSPRL |
|---|---|---|
| Sample complexity | 253.1 | 520.0 |
| Visits to trap state | 44.6 | 4.7 |

appears between minimizing the sample complexity and visiting undesirable states. This numerical simulation shows that GOSPRL can naturally adjust this trade-off by cost-weighting the successive SSP problems it tackles.