# OpenReview forum: "A Provably Efficient Sample Collection Strategy for Reinforcement Learning"
_NeurIPS.cc/2021/Conference — NeurIPS 2021 Spotlight_

### Official Review · Reviewer_2PPM · 2021-06-30

**Rating:** 8
**Confidence:** 4

**Summary:**

The paper showcases a polynomial sample complexity for the problem of simulating a generative model through online interactions with a communicating MDP. The approach works as follows: It first prescribes the samples to be taken in each state-action pair for a specific objective, and it actively collects these samples by addressing a sequence of SSP problems. The paper includes a numerical validation in illustrative domains.

**Limitations And Societal Impact:**

I think the limitations are well-discussed in the paper and the appendix, which provide details on how to relax the main assumption (communicating MDP) and how to improve the empirical performance of GOSPRL.

**Main Review:**

This work provides a simple paradigm to translate any generative-model algorithm into a provable efficient online learning procedure, effectively bridging the gap between two fundamental settings of theoretical RL.
The paper is clean, the approach and the problem are well-motivated, and the related discussion is really thorough. It also includes the analysis of a bunch of relevant objectives that can be targeted with the proposed procedure. Especially, the goal-free and cost-free exploration setting (Sect. 4.3), which generalizes the reward-free exploration framework to infinite-horizon and goal-conditioned RL, might be of independent interest, and a nice direction for future works developing objective-specific approaches.
Overall, I think that this work will provide a strong contribution to the theoretical RL field, and I recommend accepting it.
Some additional comments and questions to the authors can be found below.

SAMPLE COMPLEXITY AND THE LOWER BOUND

The paper reports a lower bound of $\Omega(BD)$ samples for simulating the generative model, and the sample complexity of GOSPRL is upper bounded by $\widetilde{O} (B D + D^{3/2} S^2 A)$. Whereas the authors note that with large sample requirements the term $\widetilde{O} (B D)$ is dominating, I am wondering under which assumptions on b(s, a) the term $\widetilde{O} (S^2 A)$ is unavoidable. Do the authors think that a careful construction might give a lower bound with an explicit dependence on $S^2 A$, which would tie the sampling problem to the one of estimating the transition model for every $(s,a)$ up to a certain confidence?

MODEL-FREE APPROACHES

The GOSPRL algorithm is inherently model-based, as it relies on empirical transitions to instantiate each SSP problem $M_k$, and the analysis is also based on concentrating empirical transitions around the real model. I am wondering if the model-based feature is really a requirement of the nature of the problem or just how the analysis is carried out. Do the authors think that it is possible to address this setting with model-free approaches?

RELATION WITH RF-RL-EXPLORE AND FUTURE DIRECTIONS

From my understanding, the GOSPRL algorithm and RF-RL-EXPLORE [30] are built on similar ideas (in a significantly different context), as they both learn policies to seek for a specific state-action pair in the inner loop. The crucial difference is that RF-RL-EXPLORE solves finite-horizon problems (with EULER) as opposed to SSP problems in GOSPRL.
Subsequent works in reward-free exploration [30, 31, 39, 64] have adopted alternative schema to improve sample complexity upper bounds. Do the authors think that similar routes could be followed for the sample collection problem?
In addition, a direct analysis of the sample complexity that GOSPRL would achieve in the (finite-horizon) reward-free exploration could draw an interesting connection between the two settings.

**Time Spent Reviewing:**

8

---

> ### Author Response · Authors · 2021-08-10
> **Response to Reviewer 2PPM**
>
> We thank the reviewer for the insightful comments. Please find our response to the questions below.
>
> **Sample complexity and the lower bound:** It would indeed be interesting to identify the conditions where an $S^2 A$ term would explicitly appear in the lower bound, which may be possible using a refined analysis. It appears challenging given current analysis tools since lower bound constructions rarely focus on lower-order terms. For example, this question also arises in the task-agnostic exploration setting [63; Wu et al., 2020] where it would be interesting to see whether there exist conditions on the task number/identity for an explicit lower-order dependence on $S^2 A$ to appear.
>
>
> **Model-free approaches:** We believe it may indeed be possible to address our problem by using a Q-learning-type approach. It would require maintaining a $Q(s,a,g)$ table, where $Q^{\star}(s,a,g)$ would denote the (unit-cost) distance from $(s,a)$ to goal state $g$ (note that the space complexity would then become similar to that of a model-based approach like GOSPRL). The algorithm could iterate as follows: (i) given the current set of under-sampled goals $G_k$ and current state $s_t$, select as goal state $g_k \in \arg\min_{g \in G_k} \min_a Q(s_t, a, g)$; and (ii) deploy a model-free strategy with goal state $g_k$. For (i), selecting as goal the "closest" undersampled state according to the current values $Q$ can be seen as the counterpart of the meta goal state that model-based GOSPRL constructs. For (ii), one may leverage a model-free strategy for regret minimization in (single-goal) SSP: in particular, the one proposed by a very recent (post-submission) work [Chen et al., 2021]. However, leveraging their analysis would lead to a “burn-in” term in our sample complexity of the order $O(\sum_{s} D_s^5 S^2 A) = O(D^5 S^3 A)$, which is much poorer than that of GOSPRL.
>
>
>
> **Relation with RF-RL-Explore and future directions:** It is not immediate how the subsequent works on reward-free exploration following [30] could be leveraged to improve our sample complexity bound. [31, 39], which optimize an adaptive upper bound on estimation errors, never explicitly lower-bound the state visitations, which is required for our setting, while [64] relies on an exploration-bonus based approach which in our case would require estimating the (unknown) diameter $D$ (to tune the bonus) and would make the “burn-in” term in the sample complexity worse than that of GOSPRL. Moreover, all these works in reward-free exploration are based on a *worst-case* analysis, although a very recent (post-submission) work [Wu et al., 2021] proposed the first problem-dependent bounds for finite-horizon reward-free exploration. We believe that a promising direction is to derive *problem-dependent* bounds in our setting, to circumvent the dependencies on $D / D_s$ which we prove to be unavoidable in the worst-case.
>
> Finally, studying what GOSPRL would achieve in finite-horizon reward-free exploration (in stationary MDPs) could indeed draw an interesting connection between the two settings. From a preliminary investigation, a relevant direction is to execute our variant GOSPRL-L (Appendix E) for increasing values of $L$, since GOSPRL-L is able to discard the sample collection at states with reachability $> L$ by tracking the cumulative time and interrupting whenever it exceeds a carefully defined threshold.
>
>
> [Wu et al., 2020] Accommodating Picky Customers: Regret Bound and Exploration Complexity for Multi-Objective Reinforcement Learning, arXiv 2020.
>
> [Chen et al., 2021]: Implicit Finite-Horizon Approximation and Efficient Optimal Algorithms for Stochastic Shortest Path, arXiv \& RL theory workshop at ICML 2021.
>
> [Wu et al., 2021] Gap-Dependent Unsupervised Exploration for Reinforcement Learning, RL theory workshop at ICML 2021.

---

> > ### Comment · Reviewer_2PPM · 2021-08-25
> > **Re: Response to Reviewer 2PPM**
> >
> > I would like to thank the authors for their thorough replies. I am still convinced this work provides a strong contribution and I will stick with my positive evaluation.

---

### Official Review · Reviewer_hnJg · 2021-07-08

**Rating:** 7
**Confidence:** 3

**Summary:**

The authors introduce a general technique for exploring the MDP in an SSP problem. Their approach extends the notion of reward-free exploration to  an "objective-agnostic" scenario, in which the goal is to generate a prescribed amount of samples from any state and action. The authors analyze the proposed algorithm and provide a sample complexity bound.

Then, the authors show this technique can be leveraged to solve specific scenarios which fit the communicating MDP setup. For these scenarios, the authors show the superiority of their approach with respect to previous works.

**Limitations And Societal Impact:**

Yes.

**Main Review:**



This paper tackles the question of how to learn the transition model up to a prescribed accuracy (in terms of amount of samples).
In terms of originality, the proposed method combines two established ideas in the RL community, reward-free exploration (in finite horizon MDPs), and exploration in the SSP problem. Therefore, the path to the analysis is quite clear and incremental in nature. Still, the problem at hand is interesting and the solution is natural and clean, so this is not necessarily a flaw. Furthermore, the authors extend the notion of reward free exploration, generalizing it to a wider set of possible tasks.
Thus, overall this paper is significant - as it provides a simple provable methodology to solve many different tasks in the SSP regime.

An interesting question regarding this approach is whether it can be interestingly extended to a case where there are several (sequential/simultaneous) tasks at hand. Indeed, in the finite horizon setting, reward-free exploration is usually analyzed to allow for a "zero-shot" solution of the MDP for any (known) reward function. Yet, in the scenario discussed in this paper, where there are no-resets and the horizon is infinite, it is worthwhile to understand what happens when there is more than one task. Do the authors have any thoughts or comments regarding this question?

Experiments: Can you add the experiment with the original treasure problem? I did not understand the comment about the "burn-in" phase, is there something fundamental here? Is it still problematic when the MDP gets bigger?

Clarity:
I enjoyed reading this paper. It is well written, the main points are well-developed and coherent and the discussions are informative and discuss the related literature properly.


Small typo:
- Lines 204,353 - "Finally,"

**Time Spent Reviewing:**

6

---

> ### Author Response · Authors · 2021-08-10
> **Response to Reviewer hnJg**
>
> Thank you for the valuable comments and suggestions. Please find our response to the questions below.
>
> **Case of sequential/simultaneous tasks:**
> This is indeed an interesting set-up that falls within the reset-free scenario that we consider.
> In the sample complexity guarantee $\widetilde{O}(BD + D^{3/2} S^2 A)$, we see that the second “burn-in” term is $B$-independent. This property of GOSPRL, combined with its model-based nature, means that it can smoothly tackle multiple tasks by skipping the “burn-in” phase, i.e., the “burn-in” term $\widetilde{O}(D^{3/2} S^2 A)$ would be independent of the number of tasks. Interestingly, we see that our goal-free cost-free exploration problem can be cast in a sequential fashion (task 1: starting from $s_0$, receive a goal $g_1$ and reach it; task 2: starting from $g_1$, receive a goal $g_2$ and reach it; … ; task $n$: starting from state $g_{n-1}$, receive a goal $g_n$ and reach it). For any $n$ and sequence of goal states $\{ g_1, …, g_n \}$ (and possibly different cost functions $\{ c_1, …, c_n \}$), instantiating GOSPRL as done in Lemma 4 yields a sequence of $\epsilon$-optimal goal-based policies. The same sample complexity guarantees as Lemma 4 would hold, since Lemma 4's guarantees are valid for *any* starting state.
>
>
> **Experiments:**
> We agree that our explanation on the Treasure experiment was poorly phrased. Since GOSPRL and our baselines (except the random policy) are all based on upper confidence bounds, they tend to display similar behaviors in the initial phases of learning, since the estimates when $n(s,a)=0$ are similar. This is the reason why we observed that the empirical performance in the Treasure-1 problem (i.e., when $b(s,a)=1$) is comparable for all learning algorithms (including the otherwise much poorer baseline of MaxEnt). As the number of samples required in each state-action increases, the difference between the algorithms' design starts making a real difference in the behavior and eventually their performance. We will clarify this point and support it with numerical simulations.

---

> > ### Comment · Reviewer_hnJg · 2021-08-17
> > **Response to response**
> >
> > I have read the response, and thank the authors for answering my questions. I stick with my original (positive) evaluation of this paper.

---

### Official Review · Reviewer_dGom · 2021-07-14

**Rating:** 7
**Confidence:** 4

**Summary:**

The paper proposes to decouple the exploration problem in RL into an objective-specific part with access to a sampling oracle, and an objective-agnostic exploration part. For the objective-agnostic part, the paper proposes the GOSPRL algorithm with bounds on its sample complexity to meet the sampling requirement. Then GOSPRL is then combined with some sampling oracle-based algorithms to provide sample complexity for three types of RL problems.

**Ethical Concerns:**

No.

**Limitations And Societal Impact:**

The settings where the proposed method works are clearly stated.

**Main Review:**

Exploration-exploitation in RL is a very challenging problem, and the paper proposes to tackle the problem by decoupling it into a objective-specific part and an objective-agnostic part. The approach is inspired by the algorithms with sampling oracles in the literature which assume to have access to sampling oracles providing samples for a given state-action pair. Such sampling oracle-based algorithms can be used for the objective-specific part, so the paper focuses on the remaining objective-agnostic exploration part.

The objective-agnostic exploration part can be viewed as an extension of the pure-exploration approach by introducing the sampling requirement function for each state-action pair. By utilizing techniques from stochastic shortest path problems, the paper proposes the GOSPRL algorithm to collect samples specified by the sampling requirement. Sample complexity bounds are provided by connecting the sample collection problem to a sequence of stochastic shortest path problems.

For certain problems when sampling oracle-based algorithms are available, combined with GOSPRL we can get some good sample complexity bounds for these exploration problems. The paper provides three examples: one is the sparse reward discovery problem where the sampling requirement is trivial, another one is model estimation without considering the reward, and the third one is on reward-free exploration which extends prior finite-horizon results. Simulations are provided to verify the theoretical analysis.



**Time Spent Reviewing:**

10 hours

---

> ### Author Response · Authors · 2021-08-10
> **Response to Reviewer dGom**
>
> We thank the reviewer for the positive review, and welcome all further comments.

---

### Official Review · Reviewer_WDCy · 2021-07-16

**Rating:** 6
**Confidence:** 3

**Summary:**

This paper considers the problem of learning to sufficiently explore an MDP in order to effectively provide a generative model that can be used for downstream learning tasks by algorithms that require a generative model. An algorithm is given that leverages SSP results in order to force exploration of certain state-actions that may be required by the arbitrary down-stream algorithm. Sample complexity upper and lower bounds are proved to demonstrate the samples required to meet the budget prescribed by the downstream algorithm. Several example tasks that require non-trivial exploration are shown to make use of the algorithm.

**Limitations And Societal Impact:**

See main review for limitations and suggestions.

**Main Review:**

Overall I think this is a nice paper that presents some interesting theoretical results via both upper and lower bounds on sample complexity. To the best of my knowledge, the algorithm seems new. The principle behind it is fairly straightforward in that it leverages SSP algorithms to do the heavy lifting but in a convenient way to properly explore to meet the budget. The example applications are definitely appreciated and it is promising to see that the algorithm yields competitive results theoretically in these settings. As a result, the work certainly seems thorough and there are some promising directions for future work, for example, by understanding how this extends in the function approximation setting. The related work seems to sufficiently distinguish this paper from others. The experiments are also fairly comprehensive and nicely complement the theory for the most part.

Despite the interesting results, my main reservation regards the applicability of those results. There seems to be a bit of a mismatch between how the problem is motivated in the beginning and how the results are demonstrated in the main content. From the introduction, the impression is that the sampling oracle developed in this paper is to be used along with a generative model algorithm in two stages to solve a particular downstream RL problem that would otherwise require an online algorithm. But to this, my question is: why not just use an online algorithm which is already fairly flexible and skip the two stages? This is the case for the treasure problem. The model estimation task doesn’t really specify a downstream task, so this seems to be in some conflict with the initial motivation of the paper too. The cost-free goal-free task seems to be the most convincing of three since no signal can be used by an online algorithm (but they could still be used for reward-free exploration). In summary, I am not entirely convinced of the general applicability of the results or the potential impact of solving this particular problem.

In the experiments, for the treasure problem, I am a bit skeptical of setting the budget to 10 as this is not exactly what was advertised in the problem statement in the theory. The algorithm is also only slightly better than UCRL and only in one domain (which is not necessarily a bad thing), but this leaves the reader wondering what the performance on the actual problem would be, burn-in phase and all.


**Time Spent Reviewing:**

5

---

> ### Author Response · Authors · 2021-08-10
> **Response to Reviewer WDCy**
>
> Thank you for the thoughtful comments and suggestions. Please find our response to the comments below.
>
> **Problem motivation and applicability of results:** We identify two main interests of our generic framework that decouples the learning process between an *objective-agnostic exploration strategy* (for sample collection) and an *objective-specific generative-model (GM)-based strategy* (for sample prescription):
>
> - It allows us to tackle in a unifying fashion a variety of tasks *without having to design a specific online algorithm for each*. Beyond the three presented in the main paper, we illustrate in App. I other applications that our framework can readily tackle (e.g., diameter estimation, best-state identification, reward estimation a.k.a. active exploration). Importantly, our approach comes with compelling theoretical performance. In some cases (e.g., model or reward estimation), our solution improves over the existing online algorithms directly tailored to this problem. In other novel problems (e.g., goal-free cost-free exploration), we readily obtain the first available guarantees. Going forward, we believe that GOSPRL can be used as a competitive off-the-shelf baseline when a new application is introduced.
>
> - It attempts to better understand (and partly bridge) the gap between the two fundamental settings of online RL and GM-based RL. For instance, the fact that GOSPRL improves over existing online approaches in some applications raises the question whether better application-specific online strategies may be devised. Furthermore, one possible usage of our results is in an intermediate setting where the agent would interact online with the environment and in some time steps be allowed some GM-based interaction "for free": such a setting would be smoothly tackled using the decoupled approach of GOSPRL.
>
>
>
> **Experimental details:** Since GOSPRL and our baselines (except the random policy) are all based on upper confidence bounds, they tend to display similar behaviors in the initial phases of learning, since the estimates when $n(s,a)=0$ are similar. This is the reason why we observed that the empirical performance in the Treasure-1 problem (i.e., when $b(s,a)=1$) is comparable for all learning algorithms (including the otherwise much poorer baseline of MaxEnt). As the number of samples required in each state-action increases, the difference between the algorithms' design starts making a real difference in the behavior and eventually their performance. We will clarify this point and support it with numerical simulations.
>
> We would like to point out that in the Treasure-10 problem, the performance of GOSPRL is consistently better than 0/1-UCRL in all the 7 environments tested (see Figure 1 and 7). Indeed we emphasize that by definition of our problem, the metric of interest is *not* the rate of increase of $\mathcal{P}_t$ over time but only the time needed to reach the $\mathcal{P}_t = 1$ line of success (we recall that $\mathcal{P}_t$ is the proportion of states that satisfy the sampling requirements at time $t$). GOSPRL’s steady increase of $\mathcal{P}_t$ coincide with the algorithm's design: it is *not* optimized to exhaust the sampling requirement state after state but rather to collect all sought-after samples as fast as possible.

---

### Decision · Program_Chairs · 2021-09-27

**Decision:**

Accept (Spotlight)

**Comment:**

This paper was well-received by the reviewers who all agreed that the paper studies an interesting problem and offers a solid solution. There were only minor concerns raised by one reviewer, but these were adequately addressed in the author response. Eventually, the reviewers all agreed that the paper should be accepted for publication at the conference.